# Information dynamics of *in silico* EEG Brain Waves: Insights into oscillations and functions

**Gustavo Menesse** [1,2¤] *, **Joaquín J. Torres** [1]

**1** Department of Electromagnetism and Physics of the Matter & Institute Carlos I for Theoretical and Computational Physics, University of Granada, Granada, Spain, **2** Departamento de Física, Facultad de Ciencias Exactas y Naturales, Universidad Nacional de Asunción, San Lorenzo, Paraguay

¤ Current address: Department of Electromagnetism and Physics of the Matter, University of Granada, Granada, Spain
* gmenesse@onsager.ugr.es

**Data Availability Statement:** All relevant data are within the manuscript and its Supporting information files. Data necessary for reproducing the results presented in the Figures are available in 10.5281/zenodo.11658278, including the code to

## Abstract

The relation between electroencephalography (EEG) rhythms, brain functions, and behavioral correlates is well-established. Some physiological mechanisms underlying rhythm generation are understood, enabling the replication of brain rhythms *in silico*. This offers a pathway to explore connections between neural oscillations and specific neuronal circuits, potentially yielding fundamental insights into the functional properties of brain waves. Information theory frameworks, such as Integrated Information Decomposition ($\Phi$-ID), relate dynamical regimes with informational properties, providing deeper insights into neuronal dynamic functions. Here, we investigate wave emergence in an excitatory/inhibitory (E/I) balanced network of *integrate and fire* neurons with short-term synaptic plasticity. This model produces a diverse range of EEG-like rhythms, from low $\delta$ waves to high-frequency oscillations. Through $\Phi$-ID, we analyze the network's information dynamics and its relation with different emergent rhythms, elucidating the system's suitability for functions such as robust information transfer, storage, and parallel operation. Furthermore, our study helps to identify regimes that may resemble pathological states due to poor informational properties and high randomness. We found, e.g., that *in silico* $\beta$ and $\delta$ waves are associated with maximum information transfer in inhibitory and excitatory neuron populations, respectively, and that the coexistence of excitatory $\theta$, $\alpha$, and $\beta$ waves is associated to information storage. Additionally, we observed that high-frequency oscillations can exhibit either high or poor informational properties, potentially shedding light on ongoing discussions regarding physiological versus pathological high-frequency oscillations. In summary, our study demonstrates that dynamical regimes with similar oscillations may exhibit vastly different information dynamics. Characterizing information dynamics within these regimes serves as a potent tool for gaining insights into the functions of complex neuronal networks. Finally, our findings suggest that the use of information dynamics in both model and experimental data analysis, could help discriminate between oscillations associated with cognitive functions and those linked to neuronal disorders.

generate the figures. The code for the simulation and PHI-ID analysis tools are available in an open software repository https://github.com/GuEMM/EEG_model.git and https://github.com/GuEMM/PhiID_Tools.git respectively.

**Funding:** This work is part of the Project of I+D+i, Spain Ref. PID2020 113681GBI00, funded by MICIU/AEI/10.13039/501100011033, (to JJT). J.J.T. also acknowledges financial support and from the Consejería de Transformación Económica, Industria, Conocimiento y Universidades, Spain, Junta de Andalucía, Spain and European Regional Development Funds, Ref. P20_00173. G.M. would like to thank the Programa Nacional de Becas de Postgrados en el Exterior "Don Carlos Antonio López" - BECAL of the Ministry of Economy and Finance of Paraguay for the financial sponsorship to pursue his doctoral studies in the Physics and Mathematics Program of the University of Granada. The funders had no role in study design, data collection and analysis, decision to publish, or preparation of the manuscript.

**Competing interests:** The authors have declared that no competing interests exist.

## Author summary

Electroencephalography (EEG) records cortical brain activity and is widely used in neuroscience for identifying cognitive states and diagnosing brain pathologies. However, the relationship between functional brain states and specific rhythms is sometimes unclear. Traditional methods combined with computational models often fail to link dynamical regimes to their possible functions. To address this, we used a computational model that generates *in silico* EEG-like signals in a neuron population. Instead of only analyzing spectral features, we focused our study on information flow in the neuron population between small groups of inhibitory and excitatory neurons during the emergence of different rhythms. We found that in some regimes, the system exhibits enhanced computational properties, with excitatory neurons maintaining parallel processing capacities, inhibitory neurons showing high robustness, or populations maximizing information transfer. In other regimes, low information flow results in more random behavior. Our work highlights the utility of informational dynamic analysis for understanding the relationship between emerging neuronal waves and functions in *in silico* neuronal populations, a fact that stimulates extending the present study to neuronal cultures and *in vivo* EEG time series.

## Introduction

Non-invasive electroencephalography (EEG) exploration on the cerebral cortex has become a relatively simple, convenient and inexpensive way of analyzing how large populations of neurons can cooperate to develop complex brain functions while variations of their synaptic relations occur [1–4]. In fact, the EEG technique allows for an easy visualization of spontaneous brain activity organized in terms of waves or "rhythms", which emerge due to the synchronization of millions of cortical neurons with main frequencies ranging from 0.5 to 35 Hz and more, defining the so-called $\delta$, $\theta$, $\alpha$, $\beta$ and $\gamma$ bands. Moreover, each one of these rhythms is loosely associated with different states of consciousness, such as deep sleep, anesthesia, coma, relax, and attention, and with different mental and cognitive brain processes [5].

The in-depth analysis of EEG time series has traditionally been shown to be a very useful tool to detect neurological disorders, such as epilepsy [6] and its association with autism spectrum disorder (ASD) [7], and it could also be useful to detect Alzheimer's disease (AD) in its early stages [8], and other brain pathologies. In particular, in the last years EEG data have regained significance thanks to the recent development of specific Machine Learning and Deep Learning techniques that use such data for high-accuracy detection and diagnosis of a broad range of such neuropathologies [9].

Thanks to advances in experimental techniques in recent decades, high-frequency activity (HGA) has gained significant interest in the neuroscience community [10]. HGA refers to all brain activity above 80 Hz, encompassing both pure oscillatory and non-oscillatory phenomena. The purely oscillatory phenomena are usually described as high-frequency oscillations (HFOs), defined as discrete EEG oscillatory events that clearly stand out from the background activity. The emergence of HFOs, e.g., presents a fascinating duality: they are associated with both epileptic seizures and high-level cognitive functions, underscoring the intricate interplay, not yet well understood, between brain rhythms and functions [10–15]. Neuronal activity in the high-frequency band (>80 Hz) is typically associated with cognitive functions (physiological activity), including broadband high $\gamma$ activity (80–150 Hz) and narrowband fast $\gamma$

oscillations (90–150 Hz) [16]. Conversely, high-frequency oscillations, including those in the fast $\gamma$ frequency band and even higher HFOs (>150 Hz), also serve as biomarkers for epileptic seizures (pathological HFOs), with classifications such as ripples (100–200 Hz) and fast ripples (200–500 Hz) commonly used in clinical diagnosis [10, 12, 17–19]. However, attempts to differentiate between physiological and pathological HFOs based on properties like frequency, spectral amplitude, and duration have led to contradictory results [20]. Recent research suggests that considering the co-occurrence of other phenomena, such as vertex waves and interictal epileptiform discharges, may enhance the precision of this differentiation [20]. Nevertheless, further investigation is necessary to fully grasp the distinction between physiological and pathological HFOs [21].

In general, handling real EEG data is a complex task that demands specialized expertise in signal analysis and noise reduction to avoid misinterpretation [11]. Therefore, modeling brain rhythms can offer a simpler approach to explore innovative EEG analysis techniques and conduct preliminary evaluations of hypotheses regarding the mechanisms and functions behind different actual brain rhythms.

Some of the earliest and simplest models of EEG data were presented by [22], using an excitatory/inhibitory network of integrated and firing neurons capable of generating alpha rhythms and statistical patterns similar to those observed in the thalamus. In recent literature, a reformulation of this model in terms of differential equations demonstrates that a variety of other rhythms and modulations between them can emerge based on a few relevant parameters. Moreover, the complex phenomenology observed in this simple model can be understood through concepts from statistical physics and dynamical systems theory, such as phase transitions, bifurcations, metastability, and stochastic resonance phenomena [23, 24].

While the dynamical systems/statistical physics approach has been shown to be very powerful in explaining aspects of complex systems behavior, it may not always provide a complete picture. Understanding the properties of a dynamical regime does not necessarily elucidate the "function" (in the biological sense) of a given dynamical phenomenon. Complementing this dynamical knowledge with the framework of "information dynamics" [25] offers a broader perspective on what the system is doing and what possible functions could be favored by a particular dynamical regime.

For instance, dynamical regimes characterized by high redundancy in information processing tend to be robust to failures and may be linked to vital functions in an organism, as observed in structurally coupled modular sensory-motor processing in the brain [26, 27]. Furthermore, measuring the transfer of information between parts of a dynamical system allows for the definition of an effective connectivity network, providing insights into the functional structure of the system [28] or even captures structural traits of neuronal cultures [29]. On the other hand, a regime characterized by high integrated information has recently been associated with the concept of criticality in the neural system [30], a condition possessing advantageous properties for efficient computing and information processing [31–33].

In recent years, the development of tools such as Local Information Dynamics (LID) [25] and Partial Information Decomposition (PID) [34] has significantly advanced our understanding of complex systems. LID focuses on studying how a complex system locally stores, transfers, and modifies information, while PID analyzes the information between $n + 1$ random variables ($n$ sources and one target) by decomposing it into three types of information "atoms": unique, redundant, and synergistic, and their combinations. Each type captures different fundamental relations between random variables.

In this context, a multi-target extension of PID called Integrated Information Decomposition ($\Phi$-ID) has been recently developed [35, 36]. $\Phi$-ID shows promise in providing general insights into the dynamics and information content of diverse dynamical systems, ranging

from cellular automata to networks of Kuramoto oscillators [37]. Other tools, such as Partial Entropy Decomposition [27, 38], have been employed to highlight the limitations of traditional strategies for studying high-order interactions in complex networks, including functional connectivity, total and dual total correlations, and more recently, the O-information measure [39]. Moreover, efforts to develop generalized information decomposition tools indicates a growing interest within the scientific community in the utility of new information theory frameworks for complex systems research [40].

These tools have also been applied to decompose functional interactions among brain regions, revealing disparities in information processing functions [26], and to investigate the possible existence of a whole world of unexplored structures in human brain data [27, 41]. Moreover, as demonstrated in our study, information dynamics analysis can uncover direct relationships between the microscopic activity of specific neuronal populations, local information processing, and emergent macroscopic phenomena. For instance, it can elucidate how various local physiological mechanisms—such as short-term synaptic plasticity, neuronal adaptation, and factors like underlying topology [42, 43], as well as the presence of high-order interactions in the system [44–46]—influence the generation of brain waves.

In the present study, we investigate the emergence of rhythms in an *in silico* neuronal system that generates oscillations akin to actual EEG data. Our approach diverges from traditional statistical physics and nonlinear dynamical perspectives, focusing instead on novel information theory techniques, as Integrated Information Decomposition (Φ-ID). Our objective is to elucidate the relationship between rhythm emergence and local information content and dynamics. We seek insights that help to distinguish rhythms based on the informational properties and dynamical regimes of the neuronal populations where they originate.

In our neuronal system, we observe various phenomena reminiscent of those typically seen in Local Field Potential (LFP) and EEG data, such as the emergence of high frequency oscillations (HFOs) [10, 15], coexistence of different rhythms [47, 48], continuous phase transitions [49], discontinuous phase transitions and hysteresis in neuronal activity [50], among others. Here, we delve into the local information dynamics of the network across different dynamical regimes associated with each of these phenomena. Our analysis demonstrates how information dynamics provides insights into the functions a system is inherently capable of performing in each regime. We particularly focus on High Frequency Oscillations (HFOs) and the emergence and coexistence of low and middle frequency waves, such as $\delta$ and $\beta$ rhythms, as compelling case studies of information dynamics analysis in dynamical networked systems.

Specifically, we discover that HFOs in distinct neuronal populations exhibit fundamentally different informational properties, even when exhibiting similar levels of neuronal activity. In regimes where excitatory HFOs have higher power and lower frequencies than inhibitory HFOs, the system demonstrates robust parallel processing capabilities (high redundancy maintaining differentiated information), suggesting functional dynamics akin to physiological regimes [14, 20, 51]. Conversely, when both excitatory and inhibitory HFOs emerge at the same high power and frequency, the system shows poor informational properties, potentially analogous to pathological states.

In addition to investigating High Frequency Oscillations (HFOs), we also explored the information dynamics of regimes where other rhythms emerge, such as $\delta$ and $\beta$ waves, which dominate the oscillations in our system within a meta-stable region. The coexistence and synchronization of $\delta$ and low $\gamma$—high $\beta$ waves have been related with fluid intelligence [52]. While $\delta$ waves are commonly associated with relaxed states and sleep, some studies have also linked them to cognitive operations. For instance, an increase in $\delta$ power is related to working memory and focused attention [53, 54]. In contrast, $\beta$ waves are associated with sensorimotor

control, motor preparation, sensory processing and amplification, as well as working memory allocation [54].

The relationships between specific informational properties at the local scale and the emergence of waves at the mesoscopic/macroscopic scale, as presented in this article, reinforce the notion that waves serve as signatures of functional behavior in neuronal systems. The characteristic association we found between the emergence of dominant waves and specific peaks in certain information processing modes reveals the potential of information dynamics in elucidating the fundamental relationships between neuronal dynamics, brain waves, and cognitive processes.

Viewed from the perspective of the physics of complex systems, our work exemplifies how information dynamics analysis, facilitated by tools such as Φ-ID, offers valuable insights into system emergent behavior. This approach captures intricate details about dynamical phases, collective phenomena and informational properties, complementing traditional methods in statistical physics and nonlinear dynamics. Thus, for example, we observed that the onset of a continuous phase transition, marked by a low-activity intermediate (LAI) transition [55], correlates with a peak in integrated information, and that a discontinuous phase transition in the inhibitory population coincides with a peak in informational differentiation. These observations provide novel insights into the connections between specific information dynamics regimes and phase transitions.

Although our findings indicate significant correlations between rhythms, informational regimes and phase transitions, drawing definitive conclusions about the informational properties of brain waves requires a comprehensive study of experimental data, which we plan to address in future work.

## Materials and methods

### Integrated Information decomposition (Φ-ID) framework

Before discussing the framework used to describe the information dynamics, we must clarify that here we are referring to the concept of information according to Shanon's classical Information Theory. In this way, information is defined as the surprise $h(X_i)$ [56] produced by observing a state $X_i$ of a random variable $X$ given a state probability distribution $\mathcal{P}(X_i)$. The average of the surprise (information) is called the Shannon entropy $H$ and can be viewed as a measure of "uncertainty" related to a random variable $X$ state. All the information quantities in this article are presented in bits.

In the context of classical Shannon Information Theory, Φ-ID is a multi-target extension of Partial Information Decomposition (PID) [34], which provides a unified framework to explore the information dynamics of a system through combinations of different *information atoms*. The Φ-ID framework proposes to capture how information in a system flows from present to future by decomposing the time-delayed mutual information (TDMI) into information atoms which, differently than PID, can manage not only multivariate sources, but also more than one multivariate target.

Frameworks like Φ-ID are interesting in the context of complex systems in general, and for neuronal systems in particular, because they can reveal emergent and higher-order interactions within the dynamics of a neuronal system [26]. It can be applied, for instance, in the decomposition of spontaneous spiking activity recorded from dissociated neural cultures to show how different modes of information processing are spatially distributed over the system [57] or, as in the case studied here, to investigate how such information processing modes emerge in different regions of the phase space of an *in silico* neuronal population model.

While TDMI captures the information that past states provide about the future and vice versa, without detailing how this information is processed and which part of the system provides it, $\Phi$-ID proposes to decompose mutual information as a sum of information atoms $I_\partial$, each capturing different modes of information dynamics and specifying which part of the system is operating in what mode.

To apply $\Phi$-ID we first need to separate our system into partitions and then study the information that the present (the state at time $t$) of each partition provides about its future (the state at time $t + \tau$), the other partitions and the whole system. In the simplest case, we have a bipartition $\mathcal{B}$ with parts $\mathbf{X}^1$ and $\mathbf{X}^2$, and we build an ordered set $\mathcal{A} = \{\{\{\mathbf{X}^1\}\{\mathbf{X}^2\}\}, \{\mathbf{X}^1\}, \{\mathbf{X}^2\}, \{\mathbf{X}^1\mathbf{X}^2\}\}$ called the redundancy lattice (more details concerning its mathematical properties are presented in [34, 58]).Each element of this lattice represents a type of PID atom; the first element $\{\{\mathbf{X}^1\}\{\mathbf{X}^2\}\}$ is at the bottom of the redundancy lattice and represents the information carried by both parts *redundantly,* usually represented as **Red**. The second and third elements are related to the information carried *uniquely* by the respective part, usually symbolized by $\mathbf{U}^1$ and $\mathbf{U}^2$. The last element is the information that is only accessible when both parts are considered together, which means *synergistic* information, and is commonly represented as **Syn**. Each of these atoms is represented as an $\alpha$ element of $\mathcal{A}$.

Now, if we consider the "time evolution" of these $\alpha \in \mathcal{A}_t$ ($\mathcal{A}_t$ is related to PID of the present state $t$), we will have another lattice with four PID atoms in the future, called $\beta$ atoms ($\beta \in \mathcal{A}_{t+\tau}$). Any $\alpha$ atom could evolve to a $\beta$ atom in the future, as shown in Fig 1. Finally, the time evolution proposed by the $\Phi$-ID decomposition is the product between both ($\mathcal{A}_t \times \mathcal{A}_{t+\tau}$), which provides a set of 16 composed atoms of $\alpha \to \beta$. Here, $\alpha \to \beta$ represents information atoms that were originally carried as an $\alpha$ atom (carried uniquely, redundantly, or

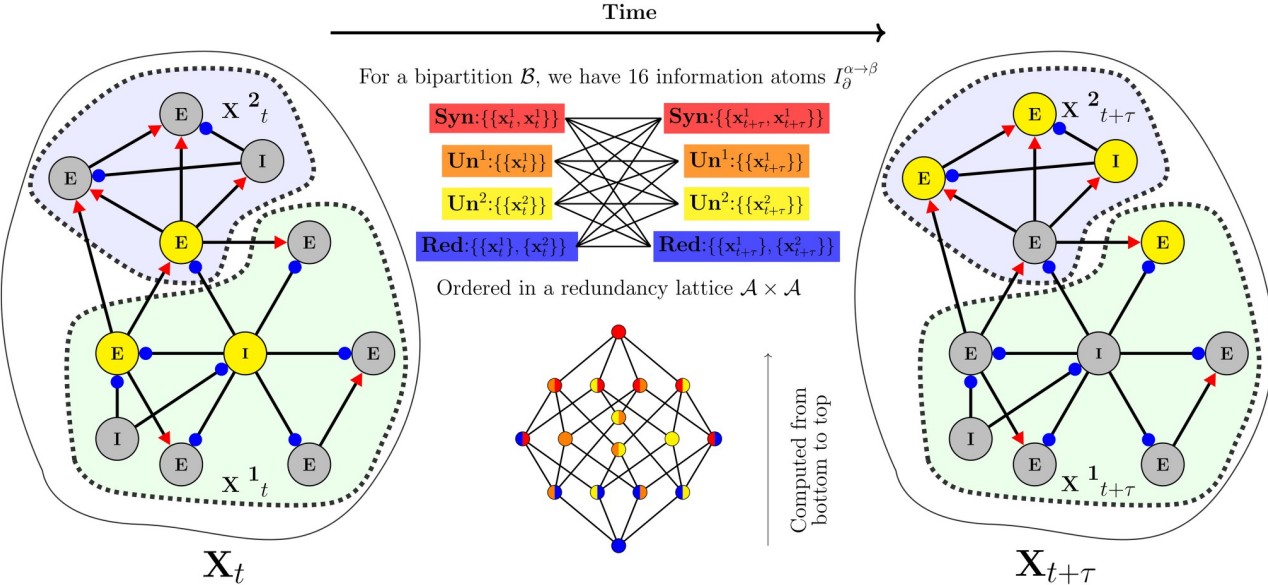

**Fig 1. Integrated Information decomposition framework ($\Phi$-ID).** The $\Phi$-ID framework describes how information is carried from present to future states of a system (information dynamics). By considering the simplest partition of a system—a bi-partition—and defining redundancy and double redundancy functions (see main text or [58]), we can compute all information atoms from the bottom to the top of the redundancy lattice (see at the bottom center of the panel). Each type of information atom captures a different mode in which information can be carried. The colors in the lattice indicate the type of atom: synergistic (red), unique (orange and yellow), and redundant (blue). Color combinations in certain atoms indicate time evolution of these atoms from one type in the present to another type in future.

synergistically in the present), which in the future will be carried as a $\beta$ atom (also carried uniquely, redundantly, or synergistically in the future). That said, mutual information is decomposed as

$$I(\mathbf{X}_t, \mathbf{X}_{t+\tau}) = \sum_{\alpha,\beta \in \mathcal{A} \times \mathcal{A}} I_{\partial}^{\alpha \to \beta} .$$ (1)

Each composed information atom $I_{\partial}^{\alpha \to \beta}$ is computed via a redundancy function, namely $I_{\cap}^{\alpha \to \beta}$, and the sum of every atom that satisfies the ordering relation $\alpha' \to \beta' \preceq \alpha \to \beta$ (see Eq (2) below). The ordering relation $\preceq$ is formally defined as $\forall \alpha, \beta \in \mathcal{A}, \alpha \preceq \beta \Leftrightarrow \forall b \in \beta, \exists a \in \alpha, a \subseteq b$. Therefore, the ordering relation of the *product lattice* $\alpha' \to \beta' \preceq \alpha \to \beta$ means that $\alpha' \preceq \alpha$ and $\beta' \preceq \beta$. In practice, this ordering relation implies that all atoms in this sum have the same or less redundant information as the computed atom (see more details [58]). Then, we can write

$$I_{\partial}^{\alpha \to \beta} = I_{\cap}^{\alpha \to \beta} - \sum_{\alpha' \to \beta' < \alpha \to \beta} I_{\partial}^{\alpha' \to \beta'}.$$ (2)

As said above, for computing the atoms, we need to define the redundancy function $I_{\cap}^{\alpha \to \beta}$. By following the axiomatic restrictions presented in the original formulation of $\Phi$-ID [58], if $\alpha = \{a_1, a_2, \ldots, a_J\}$ and $\beta = \{b_1, b_2, \ldots, b_K\}$ with $\alpha, \beta \in \mathcal{A}$ and $a_j, b_k$ non-empty subsets of $\{1, \ldots, N\}$, this function can be reduce to PID redundancies, as

$$I_{\cap}^{\alpha \to \beta} = \begin{cases} \mathbf{Red}(\mathbf{X}_t^{a_1}, \ldots, \mathbf{X}_t^{a_J}; \mathbf{X}_{t+\tau}^{b_1}) & \text{if } K = 1 \\ \mathbf{Red}(\mathbf{X}_{t+\tau}^{b_1}, \ldots, \mathbf{X}_{t+\tau}^{b_K}; \mathbf{X}_t^{a_1}) & \text{if } J = 1 \\ I(\mathbf{X}_t^{a_1}, \mathbf{X}_{t+\tau}^{b_1}) & \text{if } J = K = 1 . \end{cases}$$ (3)

This let us with the responsibility of choosing wisely a PID redundancy function **Red**. Note that last expressions are not defined when both $K$ and $J$ are greater than 1. This situation occurs in the atom of double redundancy $I_{\cap}^{\mathbf{Red} \to \mathbf{Red}}$. Therefore, we need also to define a double redundancy function. After defining both redundancy and double redundancy functions we can compute all information atoms starting from the lower atoms in the order relation.

There is no consensus on an universally preferable redundancy function, as this is still a work in progress [35, 58]. In the present work, for simplicity and to maintain a computationally tractable analysis, we only explore bipartitions of our system and use mutual minimum information (MMI) as redundancy and double redundancy functions. For a two-part system $\mathbf{X}_i$ and $\mathbf{X}_j$, we have then:

$$\mathbf{Red}(\mathbf{X}_t^{a_1}, \ldots, \mathbf{X}_t^{a_J}; \mathbf{X}_{t+\tau}^{b_1}) = \min_i I(\mathbf{X}_t^{a_i}; \mathbf{X}_{t+\tau}^{b_1}) ,$$ (4)

$$\mathbf{Red}(\mathbf{X}_{t+\tau}^{b_1}, \ldots, \mathbf{X}_{t+\tau}^{b_K}; \mathbf{X}_t^{a_1}) = \min_j I(\mathbf{X}_t^{a_1}; \mathbf{X}_{t+\tau}^{b_j}) ,$$ (5)

and

$$I_{\cap}^{\mathbf{Red} \to \mathbf{Red}} = \min_{i,j} I(X^i; X^j) .$$ (6)

Once a redundancy function is defined and after estimating the probability distributions of $t$ and $t + \tau$ states of the system, the decomposition only requires solving a system of 16 linear

equations [58], which is straightforward. A schematic representation of the framework is presented in Fig 1.

As previously mentioned, TDMI can quantify the information that the past or present state of the system provides about its future. However, it does not reveal how this information is actually transferred from past to future and vice-versa. For instance, if a system relies predominantly on the state of a specific group of neurons, the majority of the measured TDMI would be attributed to this group. Consequently, any failure or alteration within this group would significantly impact our information of the future states of the system. Alternatively, information might be redundantly provided by many groups of neurons, meaning that any single neuron or group gives the same information about the future. In this scenario, removing some neurons or groups would not substantially affect our knowledge of the future. Finally, there may be a regime where information about the future is derived only by knowing the states of all groups of neurons jointly, known as synergistic information.

In general, we can make the following interpretation about how the TDMI is decomposed. If there are many groups that carry unique information, this system is a highly differentiated system (with specialized parts). If the system is redundancy dominated, it will have a high robustness to failures as many parts of the system carry the same information. This last indicates that such parts of the system must have the same functional properties. Finally, a system that is synergistic dominant is a high-integrated system, where all parts work together to determine the future states of the system.

To identify the different dynamic information regimes, we group the atoms into 3 measures. The first was the *revised effective information* $\varphi^R$, proposed by [36, 37], which for a given bipartition $\mathcal{B}$ is defined as

$$\varphi^R[\mathbf{X}, \tau, \mathcal{B}] = I(\mathbf{X}_t; \mathbf{X}_{t+\tau}) - \sum_{j=1}^{2} I(\mathbf{X}_t^j; \mathbf{X}_{t+\tau}^j) + \min_{ij} I(\mathbf{X}_t^i; \mathbf{X}_{t+\tau}^j) \,. \tag{7}$$

From this revised effective information, the corresponding *revised integrated information* $\Phi^R$ is defined as the normalized revised effective information of the minimum information partition (MIP), i.e.:

$$\Phi^R[\mathbf{X}, \tau] \quad = \quad \varphi^R[\mathbf{X}, \tau, \mathcal{B}^{MIP}]/\mathcal{K}(\mathcal{B}^{MIP})$$

$$\mathcal{B}^{MIP} \quad = \quad \arg_{\mathcal{B}}\min \frac{\varphi^R[\mathbf{X}, \tau, \mathcal{B}]}{\mathcal{K}(\mathcal{B})} \tag{8}$$

$$\mathcal{K}(\mathcal{B}) \quad = \quad \min\{H(\mathbf{X}^1), H(\mathbf{X}^2)\} \,,$$

where $\mathcal{K}$ is a normalization factor equal to the minimum entropy between the partition entropies [59]. Note that the revised effective information $\varphi^R$ captures synergistic information plus the information that is transferred between parts of the system, while $\Phi^R$ aims to quantify the extent to which the parts of a system work together as a whole (the extent to which the whole is more than the sum of its parts).

Based on the strategy used in the revised integrated information measure shown in Eq (8), we propose in the present work two new measures to capture differentiation and redundancy in the system. First, to compute the level of differentiation of information, we measure the

information that is uniquely carried by each partition across time, which is defined as

$$Un[\mathbf{X}, \tau, \mathcal{B}] = \sum_k^2 \left[ I(\mathbf{X}_t^k; \mathbf{X}_{t+\tau}^k) - \min_i I(\mathbf{X}_t^k; \mathbf{X}_{t+\tau}^i) - \min_j I(\mathbf{X}_t^j; \mathbf{X}_{t+\tau}^k) + \min_{ij} I(\mathbf{X}_t^i; \mathbf{X}_{t+\tau}^j) \right] . \quad (9)$$

Now as before, we first normalize $Un[\mathbf{X}, \tau, \mathcal{B}]$ using $\mathcal{K}(\mathcal{B})$, and then search the partition that minimizes this quantity, which we denote as $\mathcal{B}^{MUP}$ (here MUP means minimum unique information partition)

$$\mathcal{U}[\mathbf{X}, \tau] \quad = \quad Un[\mathbf{X}, \tau, \mathcal{B}^{MUP}] / \mathcal{K}(\mathcal{B}^{MUP}) \quad (10)$$

$$\mathcal{B}^{MUP} \quad = \quad \arg_\mathcal{B} \min \frac{Un[\mathbf{X}, \tau, \mathcal{B}]}{\mathcal{K}(\mathcal{B})} . \quad (11)$$

As this unique information is conceptually the opposite of the integrated information, we will call it *differentiated information.*

Third, to quantify the degree of redundancy of our system, we define a non-synergistic redundancy (NS-*Red*) information measure adding all redundancy-related atoms that do not contain a synergistic component. This implies to subtract four times the double redundancy atom (see Eq (2)). This could imply the possibility of negative values for $\mathcal{R}$. We can avoid this by adding four times the double redundancy atom in the final definition of NS-*Red*, which results in

$$\text{NS-}Red = \min_{ij} I(\mathbf{X}_t^i; \mathbf{X}_{t+\tau}^j) + \sum_k^2 \left[ \min_i I(\mathbf{X}_t^k; \mathbf{X}_{t+\tau}^i) + \min_j I(\mathbf{X}_t^j; \mathbf{X}_{t+\tau}^k) \right] . \quad (12)$$

With this measure, we define the redundant information $\mathcal{R}$ as the minimum non-synergistic redundancy, and is defined as

$$\mathcal{R}[\mathbf{X}, \tau] \quad = \quad \text{NS-}Red[\mathbf{X}, \tau, \mathcal{B}^{MRP}] / \mathcal{K}(\mathcal{B}^{MRP}) \quad (13)$$

$$\mathcal{B}^{MRP} \quad = \quad \arg_\mathcal{B} \min \frac{\text{NS-}Red[\mathbf{X}, \tau, \mathcal{B}]}{\mathcal{K}(\mathcal{B})} . \quad (14)$$

where $\mathcal{B}^{MRP}$ is the partition that minimizes the non-synergistic redundancy (MRP meaning minimum redundant information partition).

The atoms that make up each measure are indicated in Fig 2 using the redundancy lattice representation. In this figure, we see that the revised effective information Eq (7) is constituted by a synergistic component and a transfer component. Information transfer is particularly useful for identifying dynamical regimes that maximize communication between parts of the system. Therefore, using the Φ-ID framework, we will explicitly define information transfer for a given bipartition $\mathcal{B}$ as follows:

$$\mathcal{T}[\mathbf{X}, \tau, \mathcal{B}] = \sum_i \sum_{\substack{j \\ j \neq i}} I(\mathbf{X}_t^i; \mathbf{X}_{t+\tau}^j) - \sum_k^2 \left[ \min_i I(\mathbf{X}_t^k; \mathbf{X}_{t+\tau}^i) + \min_j I(\mathbf{X}_t^j; \mathbf{X}_{t+\tau}^k) - \min_{i,j} I(\mathbf{X}_t^i; \mathbf{X}_t^j) \right] . \quad (15)$$

As we will see later, it is important to look for the partition that maximizes $\mathcal{T}$, since such maximum of the transfer information can be related, for instance, with the emergence of specific waves in both excitatory and inhibitory populations.

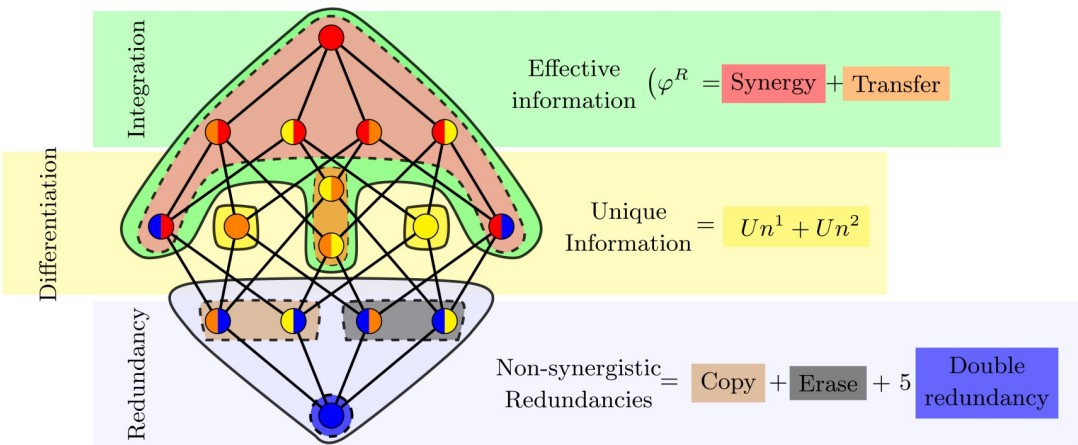

**Fig 2. Measures to capture information dynamics.** Information measures computed from information atoms presented in the redundancy lattice. Each node has two PID atoms from the present to the future ($\alpha \to \beta$) indicated by a color code: synergistic (red), unique 1 and 2 (orange and yellow) and redundant (blue). The colored area indicates the atoms that make up each measure. Revised integrated information [37] (green), unique information (yellow) and non-synergetic redundancies (blue).

Finally, we define a measure to identify the partition that minimizes information storage (IS) in our system. We use the definition of IS presented in [58], which defines it as the sum of $\Phi$-ID atoms that do not change over time. This includes the sum of $I_\partial^{\mathrm{Syn}\to\mathrm{Syn}}$, $I_\partial^{\mathrm{Un}^1\to\mathrm{Un}^1}$, $I_\partial^{\mathrm{Un}^2\to\mathrm{Un}^2}$ and $I_\partial^{\mathrm{Red}\to\mathrm{Red}}$ which gives

$$
\begin{aligned}
\mathrm{IS}[\mathbf{X}, \tau, \mathcal{B}] &= I(\mathbf{X}_t; \mathbf{X}_{t+\tau}) - \sum_k^2 \left[ I(\mathbf{X}_t^k; \mathbf{X}_{t+\tau}) + I(\mathbf{X}_t; \mathbf{X}_{t+\tau}^k) \right] + \sum_{\substack{i,j \\ i \neq j}} I(X_t^i; X_{t+\tau}^j) \\
&\quad + 2\sum_k^2 \left\{ I(X_k^t; X_k^{t+\tau}) - \min_i I(X_i^t; X_k^{t+\tau}) - \min_k I(X_k^t; X_j^{t+\tau}) \right\} \\
&\quad + \min_i I(\mathbf{X}_t^i; \mathbf{X}_{t+\tau}) + \min_j I(\mathbf{X}_t; \mathbf{X}_{t+\tau}^j) + 4 \min_{ij} I(X_t^i; X_{t+\tau}^j)
\end{aligned}
\tag{16}
$$

Then again, we propose defining "Storage" as the normalized IS of the minimum information storage partition, namely $\mathcal{B}^{MISP}$, resulting in the following mathematical definition

$$
\mathrm{Storage}[\mathbf{X}, \tau] = \mathrm{IS}[\mathbf{X}, \tau, \mathcal{B}^{MISP}] / \mathcal{K}(\mathcal{B}^{MISP})
\tag{17}
$$

$$
\mathcal{B}^{MISP} = \arg_\mathcal{B} \min \frac{\mathrm{IS}[\mathbf{X}, \tau, \mathcal{B}]}{K(\mathcal{B})} .
\tag{18}
$$

It should be noted that the redundant information ($\mathcal{R}$), differentiated information ($\mathcal{U}$), and "Storage" are new measures inspired by the integrated information measure strategy of defining an information dynamics mode by identifying bottlenecks in a given informational property of a system. With these measures, we can describe most of informational regimes of the emergent phases of the neuronal system described in the next section.

## EEG brain rhythms model

The model studied here for the generation *in silico* of brain rhythms was first introduced in [22] to reproduce EEG $\alpha$-rhythms obtained from activity recorded from the thalamus.

Recently, the model was formalized and extended in [23] and [24] to study the emergent oscillatory response of a balanced E/I neural population to variable noisy inputs, and explored the possibility of stochastic resonance phenomena and the possible relation between brain rhythm modulation and dynamic phase transitions. One of the main conclusions from these works is that the model is capable of reproducing the features of familiar brain rhythms in EEG recordings just by varying the level of noisy uncorrelated activity that arrives at the neural population from other areas [23]. Moreover, in [24] it has been demonstrated the important role of short-term synaptic plasticity to induce explosive phase transitions from "non-pathological" to "epileptic-like" oscillatory behaviour.

The model considers two coupled regular two-dimensional networks with periodic boundary conditions (lattice on a torus). The first one, with $N_E$ excitatory neurons disposed in a square lattice with $c_E$ rows/columns and the second one with $N_I$ inhibitory neurons disposed also in a square lattice with $c_I = \frac{c_E}{2}$ columns; in this way for each pair of excitatory columns an inhibitory column fits in between, as shown in Fig 3. This model aims to capture the essentials of the cerebral cortex, where excitatory neurons are reported to occur almost four times more frequently than inhibitory neurons [60], a ratio which is assumed to correspond to a balanced state of the cortex.

To coupling both lattices, each inhibitory neuron receives input from spikes generated by 32 nearby presynaptic excitatory neurons and then sends spikes to 12 adjacent postsynaptic excitatory neurons, as shown in Fig 3. Following previous works [22–24], we only consider excitatory-inhibitory (E-I) and inhibitory-excitatory (I-E) interactions, as including excitatory-excitatory (E-E) and inhibitory-inhibitory (I-I) connections does not significantly alter the dynamical phase space or the emergent oscillatory behavior in the network [24].

Using the leaky integrated and fire neuron dynamics to monitor the time dependence of the membrane potential of each neuron, we describe the dynamics of the excitatory or

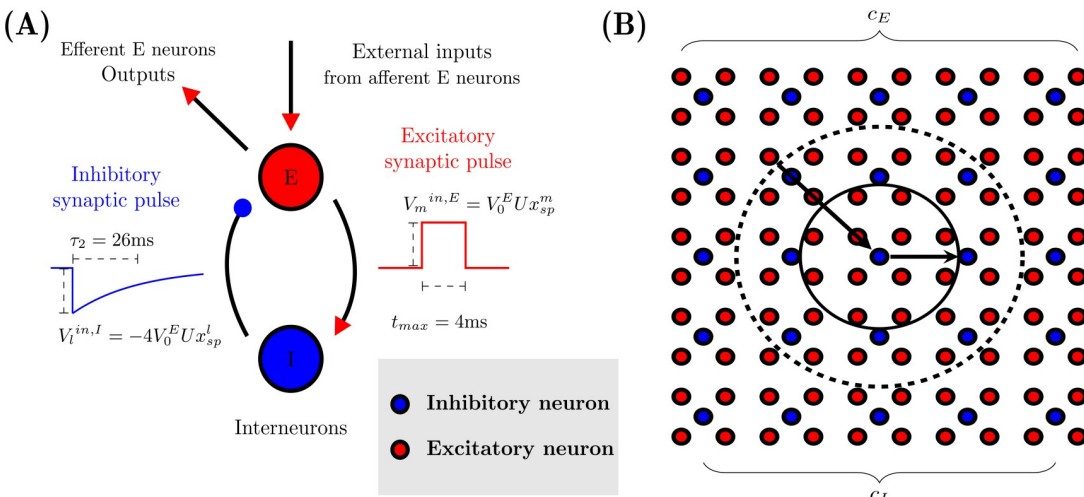

**Fig 3. Model for generation *in silico* of EEG brain rhythms: Local dynamics and network topology.** (A) The model implements a minimal neuronal circuit, where excitatory and inhibitory pulses are different and act in different timescales. (B) Schemes representing the network topology used in the present study as in [22–24]. Inhibitory neurons (blue dots) have 32 presynaptic excitatory neighbours (red dots inside the dashed circle) while each of them are pre-synaptic neighbours of 12 excitatory neurons (inside the solid line circle).

inhibitory membrane potential $V^{E/I}$ as

$$\tau_m \frac{dV_i^E(t)}{dt} \;=\; V_r - V_i^E(t) + \frac{V_{min} - V_i^E(t)}{V_{min}} \sum_{m=1}^{K_I=3} V_{m=1}^{in,I}(t) + V^{noise}(t) \tag{19}$$

$$\tau_m \frac{dV_i^I(t)}{dt} \;=\; V_r - V_i^I(t) + \frac{V_{sat} - V_i^I(t)}{V_{sat}} \sum_{l=1}^{K_E=32} V_l^{in,E}(t) \,, \tag{20}$$

where the factors $\frac{V_{sat} - V}{v_{sat}}$ and $\frac{V_{min} - V}{V_{min}}$ in the input terms were introduced to prevent physiologically unrealistic levels of membrane potential (too high or too low) around the resting membrane potential that it is set to $V_r = 0$. The limits used were $V_{sat} = 90$ mV and $V_{min} = -20$ mV. Furthermore, the time constant $\tau_m$ is equal to $\tau_1(\tau_2)$ depending if the membrane cell voltage is above(below) the resting potential $V_r$ [22]. The terms $V_\sigma^{in,I/E} \quad \sigma = m, l$ correspond to synaptic inputs from a neighboring presynaptic neuron when a spike occurs. These inputs vary depending on the nature of the spike (excitatory or inhibitory) and follow the equations,

$$V_m^{in,I} \;=\; V_0^I U x_{t_{sp}} \Theta\left(t - t_{sp}\right) e^{-\frac{t - t_{sp}}{\tau_2}} \tag{21}$$

$$V_l^{in,E} \;=\; V_0^E U x_{t_{sp}} [\Theta(t - t_{sp}) - \Theta(t - t_{sp} - t_{max})] \,, \tag{22}$$

where $V_0^{E/I}$ is the maximum amplitude of synaptic input, $U$ is the release probability of neurotransmitter vesicles (viewed also as the fraction of resources released) and $x_{t_{sp}} = x(t_{sp})$ is the fraction of neurotransmitters available (i.e. which can be released) after the arrival of an action potential at time $t_{sp}$ [61] (see below). In this model, excitatory spikes generate synaptic inputs in the form of square pulses of duration $t_{max}$, while inhibitory inputs generate pulses with exponential decay with decay time constant $\tau_2$ [22].

The system is also driven by a noise term $V^{noise}$ which accounts for the excitatory inputs to E neurons from neurons in other regions of the brain. This noise is modeled assuming the lack of temporal correlations using a Poisson signal characterized by a noise level parameter $\mu$. This represents the mean value of external action potentials reaching each E neuron in 100 simulation time steps, which means that an E neuron receives on average $\lambda = \frac{\mu}{100}$ external spikes at each time step. In the simulations, we assume the existence of $n$ external neurons so that the probability that each E neuron will receive an external spike from one external neuron is $\lambda/n$. Using a sufficiently large $n$, the external input per unit of time follows a Poissonian distribution. For all the simulations in this paper, we set $n = 100$.

We also account for the possibility of synaptic plasticity at the synapses and consider a simple mechanism of short-term depression (STD) in which synaptic efficacy, represented by the fraction of available neurotransmitters $x(t)$, decreases with the increase in the presynaptic firing rate. This is due to the rapid depletion of neurotransmitters inside the synaptic button and their slow recovery after heavy presynaptic activity [62]. It has been demonstrated that this STD mechanism has strong computational implications in the functioning of different neural systems [63] such as an increase in memory capacity [64], optimal transmission of information in noisy environments [65, 66], appearance of dynamical memories [67], and even it could be a mechanism to avoid over synchronization [68]. The STD mechanism is introduced by the

equation

$$\frac{dx(t)}{dt} = \frac{1 - x(t)}{\tau_{rec}} - Ux(t)\delta\left(t - t_{sp}\right),$$ (23)

with the delta function indicating that the second right-hand term occurs only for $t = t_{sp}$. For simplicity, concurrent mechanisms such as short-term synaptic facilitation [69] were not included.

The set of Eqs (19–23) describes the dynamics of the neuron membrane potential below a firing threshold, namely $V_{th}$, so when $V_i^{E/I} > V_{th}$ a spike is generated. Following previous work, to introduce the possibility of absolute ($t_a$) and relative refractory periods after the generation of a voltage spike at $t_{sp}$, the firing threshold is considered to be time dependent following the evolution

$$V_{th}(t) = \begin{cases} V_{sat} & t_{sp} < t < t_{sp} + t_a \\ V_{th}^0 + (V_{sat} - V_{th}^0)e^{-\kappa(t - t_{sp} - t_a)} & t > t_{sp} + t_a \end{cases}.$$ (24)

Here, when a spike takes place, the threshold is first set to $V_{sat}$ during a period of $t_a$ ms to prevent any further spike generation accounting in this way for an absolute refractory period. Then, it decays exponentially to its resting value $V_{th}^0$ with a time constant $\kappa^{-1}$ that mimics the existence of a relative refractory period.

All parameters, symbols and values used in our study are summarized in Table 1. Throughout the text, we use the noise level $\mu$ and the recovery time constant $\tau_{rec}$ as the system control parameters, while all the other parameters were fixed. All differential equations in the model were numerically integrated using a simple first-order Euler method with time step $\Delta t = 4/100$ms.

We have explored the dynamic phase space of the model by simulating $10^7$ time steps for different values of $\tau_{rec}$ and $\mu$, reproducing the results presented in [24] and obtaining the different dynamical regimes and phase transitions already reported. In all simulations, we used a network with $c_E = 14$ ($N_E = 196$) and $c_I = 7$ ($N_I = 49$). To monitor the emerging rhythms in the

**Table 1. Meaning, symbols and values of the EEG model parameters.**

| Parameters | Symbols | Values |
|---|---|---|
| Depolarized membrane potential time constant | $\tau_1$ | 16 ms |
| Hyper-polarized membrane potential time constant | $\tau_2$ | 26 ms |
| Synaptic resource recovery time constant | $\tau_{rec}$ | 0 to 300 ms |
| Proportion of synaptic resources released at each spike | $U$ | 0.5 |
| Maximum excitatory input amplitude | $V_0^E$ | 10 mV |
| Maximum inhibitory input amplitude | $V_0^I$ | −40 mV |
| Resting membrane potential | $V_r$ | 0 mV |
| Saturation membrane potential | $V_{sat}$ | 90 mV |
| Minimum allowed membrane potential | $V_{min}$ | −20 mV |
| Excitatory pulse duration | $t_{max}$ | 4 ms |
| Firing membrane potential threshold | $V_{th}^0$ | 6 mV |
| Absolute refractory period | $t_a$ | 4 ms |
| Relative refractory period time constant | $\kappa^{-1}$ | 0.5 ms |
| Number of external neurons | $n$ | 100 |
| Noise level | $\mu$ | 0 to 15 |
| Integration time step | $\Delta t$ | 0.04 ms |

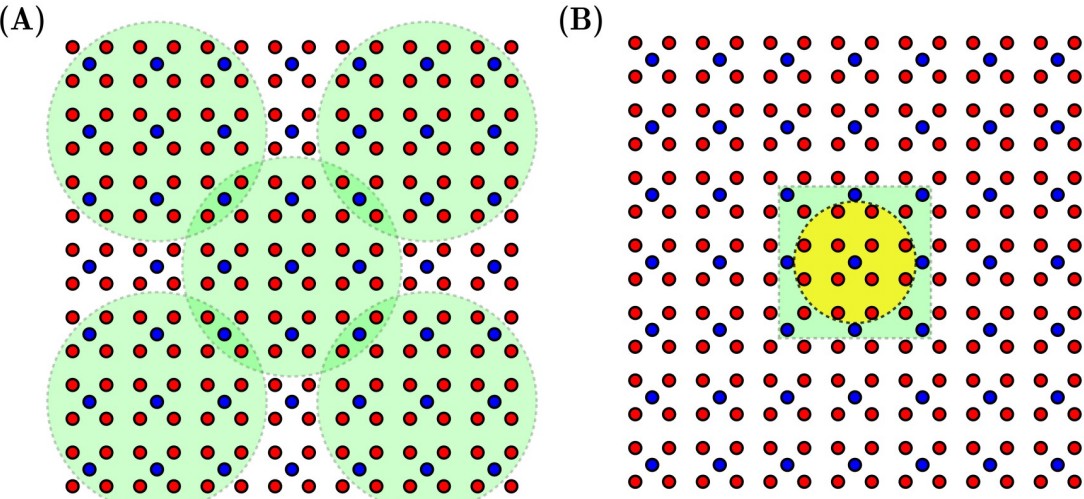

**Fig 4. Measurement schemes used for the study of the information dynamics of the network.** (A) LPF-scheme: Five groups of 32 E and 9 I neurons (inside green circles) across the network were selected. The averaged membrane potential for each group was calculated, generating a five-channel EEG-like time series. (B) Scheme for Φ-ID analysis: 12 E neurons (E neurons inside the yellow circle) and 9 I (I neurons inside the green square) were selected. The states (active/inactive) of each of them were saved to obtain time series of discrete Boolean variables.

system, we define a local field potential (LFP) scheme, which consists of measuring the averaged membrane potential of the network by segmenting the network into 5 regions, as shown in Fig 4A. More precisely, We selected groups of 32 E and 9 I neurons and then computed the averaged membrane potential at each time step over the whole group of neurons, obtaining in this way and separately a kind of LFP time series for excitatory and inhibitory neurons.

At fixed parameters $\tau_{rec}$ and $\mu$, the obtained averaged membrane potential over the neuron group (or LFP) are expected to be more or less stationary as the system reaches its stationary regime; therefore Fourier transforms should be enough to analyze the power spectrum of each LFP time series. Furthermore, taking LPF measures of excitatory and inhibitory populations separately allows us to explore the differences between rhythms in each population, analysis that will not be possible in experimental data.

Beyond the membrane potential, the time series of the spike trains for each neuron was also calculated by saving the state $X_i$ of each neuron. The state is a Boolean variable $X_i = \{0, 1\}$, with 0 for inactive or silent neurons, and 1 for active or firing neurons. Time binning was also applied to reduce the size of the time series and reduce the computational resources needed for the analysis. Considering that the absolute refractory period in simulations is set to 100 time steps (4ms at $\Delta t = 4/100$ ms), we used bin widths of 100 time steps. If a neuron spikes in that interval, the $X_i$ has value 1, otherwise its value was 0. These spike trains were used to study the information dynamics of the networks, as discrete variables are simpler to analyze with information theory tools that relates on probability and/or entropy estimators.

To explore the whole phase space, and following the previous literature, we used the noise level $\mu$ and the recovery time $\tau_{rec}$ as relevant control parameters. On the other hand, we use the time average of the excitatory and inhibitory network activity as order parameters. These are defined, respectively, as the proportion of active neurons (firing neurons) in a given time bin $t$ in each neuron population, i.e. $\rho^E[t] = \frac{1}{N_E} \sum_i^{N_E} X_i[t]^E$ and $\rho^I[t] = \frac{1}{N_I} \sum_i^{N_I} X_i[t]^I$, where $X_i[t]$ is the Boolean state of a neuron in the time interval $t$ (being 1 when the neuron fires in that time

interval or 0 otherwise). Simulations of 800 seconds of network evolution were performed ($2 \times 10^7$ time steps), which means that $2 \times 10^5$ time bins of 4 ms were obtained to compute the network activity statistics at each point in the space of explored parameters ($\mu$, $\tau_{rec}$).

To study the information dynamics of the model, we selected the central group of 12 E and 9 I neurons (as shown in Fig 4B) and saved the individual spike train of a group of neurons; this analysis in some sense considers only local interactions (interactions between first neighbours). It is worth mentioning that the topology used here does not facilitate the presence of long-range interactions, so observing an active information dynamics between distant neurons is less probable; nevertheless, it is an interesting complementary analysis to be explored in a future work.

From the spike time series, we compute the neuron state probability distribution by counting the frequency of each neuron state, first for the joint state at time $t$ and $t + \tau$ of the E and I neuron groups, $\mathbf{X}^E = \{x_1^E, \ldots, x_{12}^E\}$ and $\mathbf{X}^I = \{x_1^I, \ldots, x_9^I\}$, and second, for each possible bipartition of each group, $\mathbf{X}^{E,1} = \{x_1^E, \ldots, x_s^E\}$ and $\mathbf{X}^{E,2} = \{x_{s+1}^E, \ldots, x_{12}^E\}$ for the E group, and $\mathbf{X}^{I,1} = \{x_1^I, \ldots, x_p^I\}$ and $\mathbf{X}^{I,2} = \{x_{p+1}^I, \ldots, x_9^I\}$ for the I group. From these distributions, it was possible to compute the redundancies and, consequently, all other information atoms. Once we have calculated the information atoms for each possible bipartition of our system, we can compute the revised integrated information $\Phi^R$, the differentiated information $\mathcal{U}$, the non-synergistic redundant information $\mathcal{R}$ and other measures, as explained in the previous section.

## Results

The results section is organized as follows: First, we present the phase diagram of the system, with neuronal activity $\rho$ (see Materials and methods) serving as the order parameter, indicating the different dynamical phases and transitions lines in the model. Second, we delineate the regions within the phase diagram where different rhythms emerge, identifying specific points where each rhythm dominates. Third, we expose the behaviour of various information measures along the whole phase diagram, illustrating the relationships between informational properties and phase transitions. Additionally, we focus on the insights gained from analyzing local information dynamics regarding system behavior during rhythm emergence. Specifically we report results concerning (*i*) how information dynamics is related to the emergence of dominant $\beta$ and $\delta$ rhythms in each population, (*ii*) the informational properties of regions where there is coexistence of rhythms, and (*iii*) how information dynamics can discriminate regimes where HFO rhythms emerge in both E and I neuronal populations.

### Phase diagram, rhythms and information dynamics

**Excitatory and inhibitory activity.** The phase diagram based on the neuronal activity $\rho$ of the system in the ($\mu$, $\tau_{rec}$) parameter space is shown in Fig 5A. Three distinct regions are observed. First, for very low noise $\mu$, to the left of the continuous phase transition (white solid line), we have a subcritical regime where both excitatory (E) and inhibitory (I) neuron populations are inactive ($\rho^{E/I} \approx 0$; black region I). The second region (II), between the continuous phase transition and a discontinuous transition (white dashed line), is characterized by both E and I neuronal populations being active, but in different regimes: a low-intermediate activity regime (II.a) and a high-activity regime (II.b). Finally, a third region (region III) is observed beyond the discontinuous transition, where excitatory activity $\rho^E$ increases dramatically while inhibitory activity $\rho^I$ diminishes, as the synaptic recovery time constant $\tau_{rec}$ becomes too large to allow effective coupling between excitatory and inhibitory neurons. Between phases II.a and

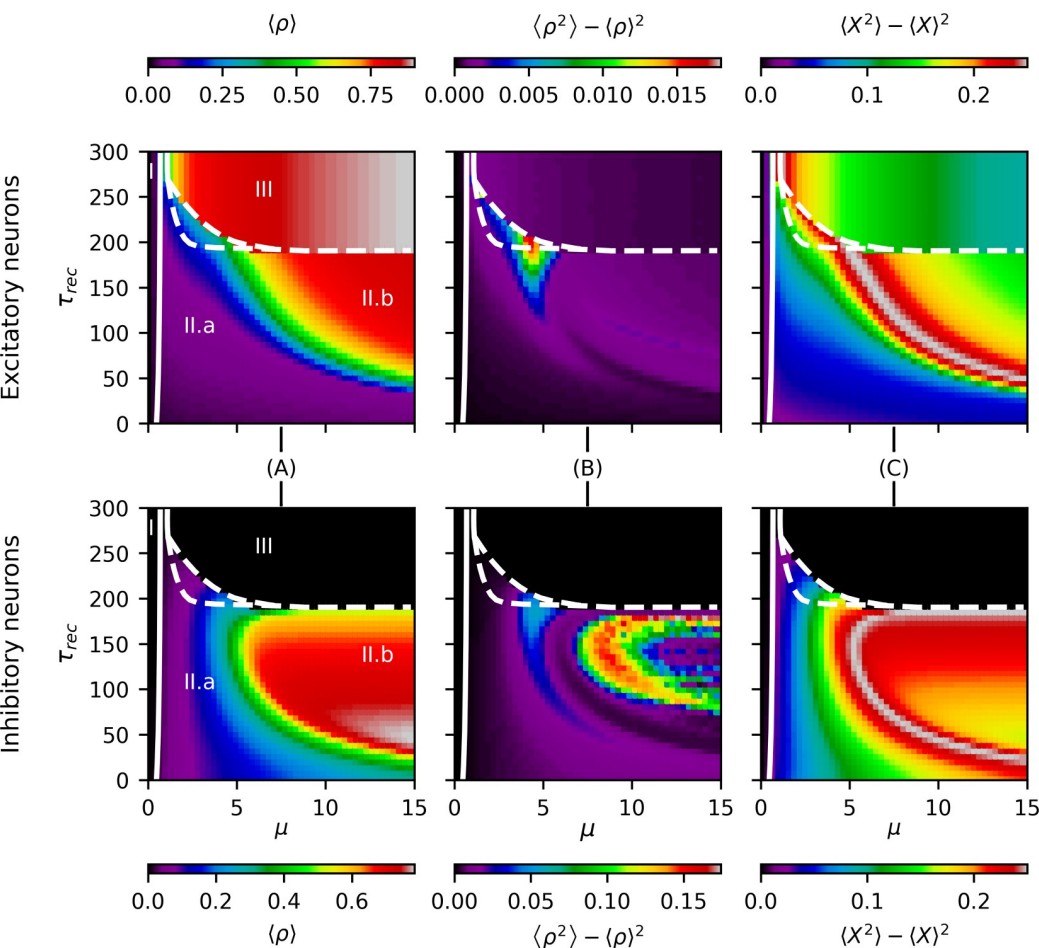

**Fig 5. Neuronal activity ρ features across the (μ, τ_rec) parameter space.** The white solid line on the left shows a second-order phase transition from a silent to an active state (AT). The white dashed line shows the emergence of an explosive first-order transition. The region between dashed lines is a meta-stable phase, as described in [24]. (A) Temporal average of excitatory and inhibitory neuronal activity $\langle\rho\rangle$: The region between the second-order and first-order transitions shows what appears to be a low activity intermediate phase (II.a), which gives place to a full active phase of high activity (II.b) for noise values $\mu > 5$. (B) Temporal variance of excitatory and inhibitory neuronal activity: The inhibitory population shows complex activity patterns along the phase space with clear regions of high temporal variability of its activity (in phase II.b), whereas the excitatory population has, in general, low temporal variability in its activity along the phase space. (C) Temporal variance of Boolean states ($X$) of neurons: The maximum variance (gray region) appears to mark the transition line between the low activity intermediate phase II.a and the high activity phase II.b. Note that this finding occurs for both the excitatory and inhibitory neuronal populations.

III, there is a meta-stable region, previously identified in this model [24], which we indicate here as the area between both white dashed lines.

We also measured the temporal variance of the activity $\rho$ of both neuronal populations (Fig 5B). We found that excitatory populations have low variability in their activity, except for a small region at the end of the metastable phase, whereas inhibitory populations exhibit rich behavior in the high-activity region (II.b). This difference could be explained by the fact that excitatory activity $\rho^E$ is driven by external noise and modulated/stabilized by inhibitory activity $\rho^I$, while this last depends solely on excitatory activity and lacks modulation, as I-I connections were not included. In future work, we will explore the impact of I-I and E-E connections on the activity fluctuations and on the information dynamics of this model.

Additionally, in Fig 5C, we illustrate the time variance of the neuron states $X$ in stationary conditions. A maximum (gray region) occurs at the transition from the low activity region (II. a) to the high activity region (II.b), suggesting the presence of a phase transition, analogous to the so-called *low activity intermediate* (LAI) phase transition [55]. This transition can be understood as a distortion of the silent-active transition between phases I and III. When inhibitory activity $\rho^I$ is introduced in the neuronal network, once the silent phase loses stability and the absorbing-active phase transition takes place (transition between I and II), the inhibitory population modulates the overall network activity, delaying the full transition to a high activity state (phase II.b). This behavior has been previously described in simpler neuronal networks with sparse topology [55, 70, 71].

In conclusion, from this analysis, we observe four characteristic regimes associated to the network activity $\rho$:

- I) Silent phase: An absorbing or silent regime for low external drive $\mu$, which is almost independent of the time constant of the synaptic resource recovery $\tau_{rec}$.

- II.a) LAI phase: A low activity intermediate region between the second-order and first-order transition lines, occurring for small to intermediate values of $\mu$ and $\tau_{rec}$ in both neuronal populations.

- II.b) High E/I activity phase: A high activity region for both neuronal populations under high external noise $\mu$ and intermediate values of $\tau_{rec}$ ($50 \lesssim \tau_{rec} \leq 200$).

- III) Pure excitatory phase: Beyond a critical value of $\tau_{rec}$, the activity $\rho^I$ of the inhibitory population vanishes, leading to high excitatory activity driven by noise.

Next, we will explore the emergence of rhythms in the averaged membrane potential of excitatory and inhibitory neurons and relate them to the emerging phases described above.

**Dominant rhythms for relevant regions of the phase diagram.**   The relation between specific rhythms and system dynamical phases can be important for understanding how particular rhythms in the brain can emerge in terms of some physiological information (for example, level of synaptic plasticity in some brain areas in our case) and therefore to relate relevant physiological parameters with cognitive functions associated with those rhythms. With this aim, we track the main peaks of the power spectrum density (PSD) of the system's averaged membrane potential time series in different frequency bands, across the space of the relevant parameters considered. The band classification we used was the following: $(0.5 - 3.5)$Hz for $\delta$, $(7.5 - 12.5)$Hz for $\alpha$, $(12.5 - 30.5)$Hz for $\beta$, $(30.5 - 60.5)$Hz for $\gamma_{Low}$ and $> 60.5$Hz for $\gamma_{fast}$. Fig 6 depicts a phase diagram indicating the areas where the different rhythms appear in both excitatory and inhibitory populations.

The averaged membrane potential time series for each group of neurons at points of maximum power in different bands (colored stars in Fig 6) is shown in the S1 Appendix (see Fig B in S1 Appendix). Additionally, we computed the power spectrum of both excitatory and inhibitory averaged membrane potential time series at fixed values of $\mu$ and $\tau_{rec}$ (marked by black arrows in Fig 6A), which are presented in Figs C and D in S1 Appendix.

Fig 6 shows that low frequency rhythms emerge in a broad interval of synaptic recovery time scales but in a non-trivial manner. For example, as described in [24], $\delta - \beta$ modulations emerge in the meta-stable region (between phases II.a and III), but we also observe $\delta$ rhythms coexisting with other rhythms for low $\tau_{rec}$ depending on the intensity of external noise $\mu$. In the high E/I neuronal activity phase (II.b of Fig 5), we observe the emergence of excitatory and inhibitory HFOs ($\gamma_{fast}$). These regions are shown as red areas in Fig 6. The white areas indicate regions where the power spectrum does not surpass the thresholds established to define a

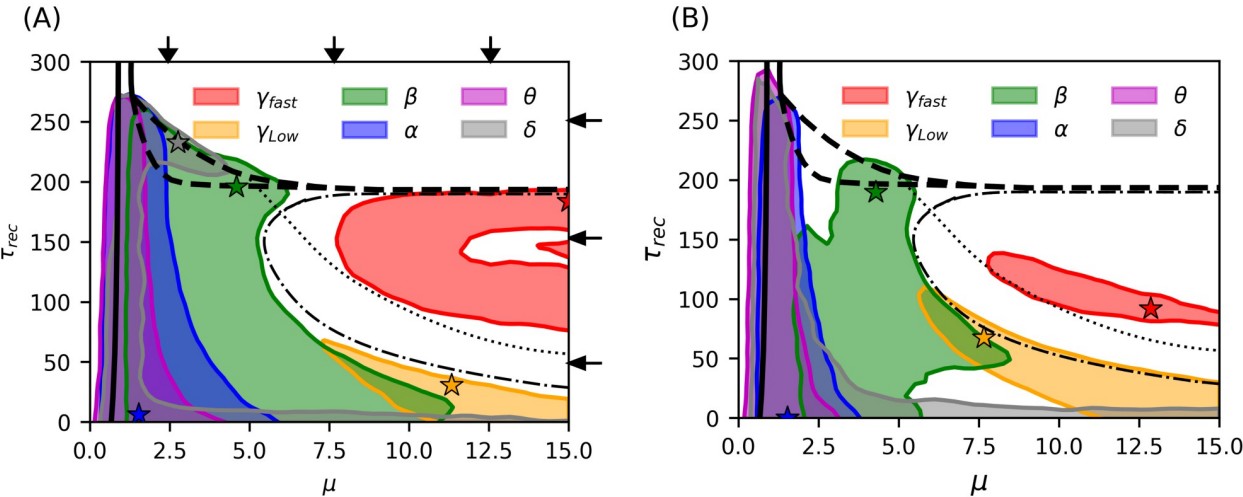

**Fig 6. Phase diagram of emerging rhythms in the model.** Rhythms observed in the (A) excitatory and (B) inhibitory neuronal population across the phase diagram. Each color shows a specific band of brain rhythms. The regions were defined as areas in the parameter space where the maximum PSD in a given band exceeds $10^{11}$, except for $\gamma_{Low}$ where the threshold is $10^{10}$. Additionally, for the inhibitory case, the threshold for $\delta$ waves is $2 \times 10^{10}$. In the white regions, maximum PSD values in each band are below the corresponding thresholds, which means there is not a clear dominant rhythm. The colored stars indicate the point of highest maximum power for each rhythm. The detailed PSD for each band is presented in Fig A in S1 Appendix. The solid line indicates a second-order phase transition (absorbing-active phase transition); dashed line indicates a first-order phase transition. Dotted and dash-dotted lines indicate the onset of transition to a LAI phase, for the excitatory and inhibitory population respectively. Vertical and horizontal black arrows indicate the values of $\mu$ and $\tau_{rec}$ for which full spectrum is shown in Figs C and D in S1 Appendix respectively.

dominant rhythm. The threshold used were $10^{11}$ for bands from $\delta$ to $\beta$ (in phase II.a) and $\gamma_{fast}$ (phase II.b) and $10^{10}$ for $\gamma_{Low}$, which is normally observed to have lower power in this model.

The emergence of high frequency oscillations ($\gamma_{fast}$), as well as other oscillations such as $\beta$ and $\delta$ waves, will be the focus of analysis in the last results subsection. Next, we will broadly explore the information dynamics across the phase diagram.

**Phase diagram and information measures.** In the present model, we used the spiking trains of excitatory and inhibitory neurons to compute various information measures across the parameter space ($\mu, \tau_{rec}$) as described earlier (see Model and Methods section). Using these information measures, we generated the phase diagrams presented in Fig 7, which include the previously discussed phase transitions (indicated by white solid and dashed lines).

Each of the information measures exhibits different behaviors across the parameter space and among neuron populations. In the excitatory population, we observe a clear peak in $\Phi^R$ at the continuous (or second-order) phase transition between phase I and III, and in the LAI phase transition between II.a and II.b (see Fig 7A top). This peak indicates a relationship with critical behavior in the network (see Discussion section). In the inhibitory population, within phase II.a, we observe that integrated information is higher near the LAI phase transition (dotted white line in Fig 7A bottom). In this region, $\beta$ and $\gamma_{Low}$ rhythms emerge in both populations. However, only in the inhibitory population we observe a relation between higher values of $\Phi^R$ and this middle frequency waves (see Fig 6B), suggesting that these intermediate frequency rhythms in the inhibitory population may represent a different phenomenon compared to the same rhythms in the excitatory population.

In region II.a, where low-frequency rhythms emerge in the excitatory population, information is carried redundantly for large values of $\tau_{rec}$ (see Fig 7B top). These results suggest that regions dominated by low-frequency oscillations ($\delta$, $\theta$, and $\alpha$) are associated with redundancy

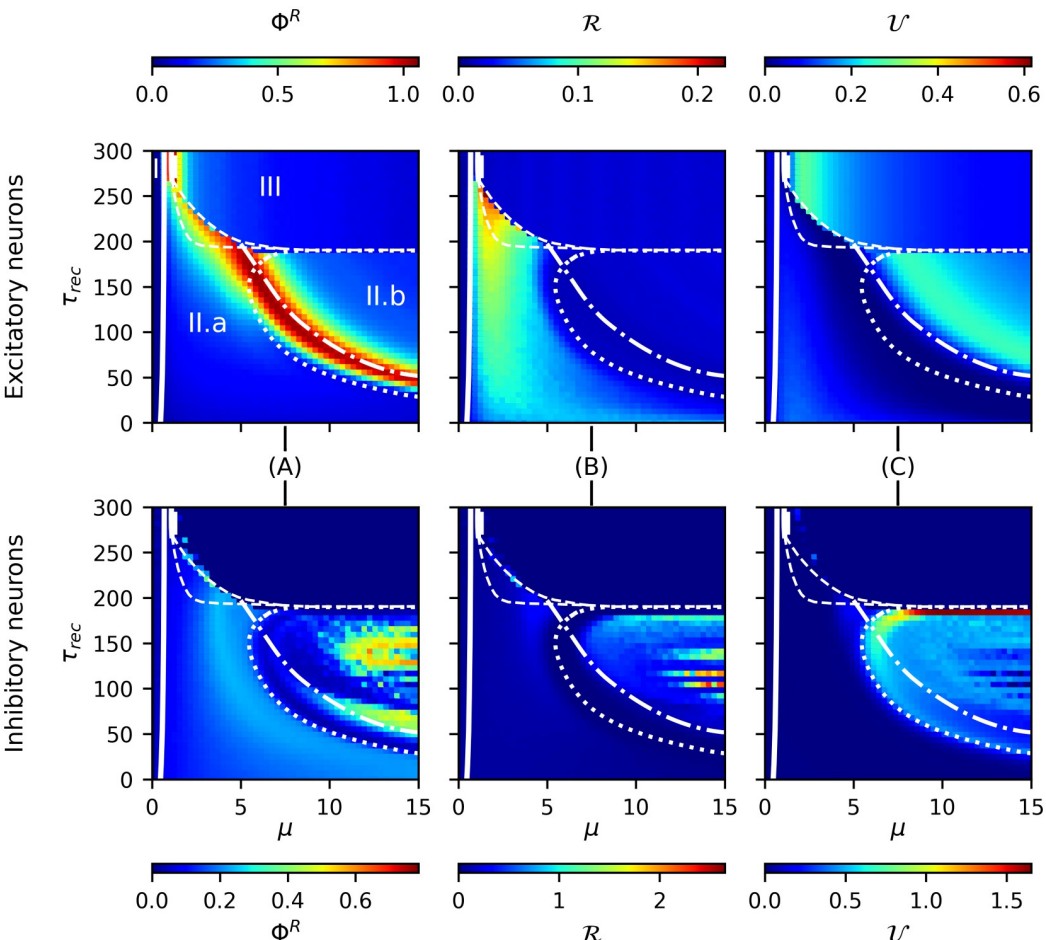

**Fig 7. Information dynamics across phase diagram for time delay τ = 1 bin (4ms).** The white solid line represents the previously explained second-order phase transition, while the white dashed line indicates the first-order phase transition present in the system. The white dotted (dash-dotted) line indicates the maximum variance in the states of the I(E) neurons (see Fig 5). The color code indicates the values of information measures. (A) Integrated information $\Phi^R$, (B) Redundant information $\mathcal{R}$ and (C) Differentiated information $\mathcal{U}$ in excitatory and inhibitory groups respectively. The neuron groups consists of 12 E neurons and 9 I neurons as indicated in Fig 3B of Material and methods section.

in the excitatory population and, as we will see in the next subsection, are also linked to information storage. Since redundancy may indicate robustness to failure, and assuming the model captures fundamental properties of brain oscillations, low-frequency waves could be associated to basal brain functions, which require greater robustness [26]. An in-depth analysis of the informational properties of the system during the emergence of these rhythms is presented in the following sections.

In Fig 7C (bottom), we observe that differentiated information exhibits an abrupt peak in the inhibitory population along the discontinuous (or first-order) transition line from region II.b to region III. This observation suggests that the explosive phase transition is associated with a corresponding information dynamics transition. Identifying specific informational properties linked to such explosive phase transition could pave the way for exploring the functional capabilities or dysfunctional behaviour of neuronal systems at the edge of a discontinuous phase transition.

In Figs H and I in S1 Appendix, we present similar phase diagrams of information dynamics for $\Phi^R$, $\mathcal{R}$ and $\mathcal{U}$ with other values of time delay, namely $\tau = 10$ bins and $\tau = 100$ bins. While in the excitatory group we see that the maximum integrated information $\Phi^R$ is invariant with the explored time delays, the other two information measures, i.e. redundancy and differentiation, decrease as the time delay increases. This indicates that excitatory information dynamics mainly involve short time scales, with the exception of the excitatory LAI phase transition. In the inhibitory population, however, both measures remain large but change in a complex manner with time delay, indicating a complex information dynamics between inhibitory neurons that spans over a larger time scale. This result reinforces the fact that inter-neurons (even in this simple model) have distinguishable information processing capabilities [72], which makes them important for understanding neuronal circuit functions, but also dysfunctional behaviour [73].

## Insights into wave emergence through information dynamics

In addition to the various phase transitions occurring in this model, an intriguing property is the emergence of low, medium, and high-frequency oscillations in both populations in a complex manner. Building on our understanding of neuronal activity and the emergence of rhythms (see previous sections), our main goal in this section is to understand the relationship between rhythms emergence and information dynamics. Specifically, we aim to characterize the informational properties of the system in regions of the parameter space where $\delta$, $\beta$, and $\gamma_{fast}$ oscillations emerge.

**Information dynamics of $\beta$ waves and rhythms coexistence regimes.** In Fig 7A, it is observed that $\Phi^R$ in the inhibitory population reaches higher values in Phase II.a close to the transition to Phase II.b (white dotted line). As presented in Fig 2, the effective information, which constitutes the backbone of the Revised Integrated information $\Phi^R$ measure, have two main components, "Synergy" and "Transfer" of information. Exploring information transfer as defined in Eq (15), we found that between inhibitory neurons, this measure reaches maximum values in the meta-stable region (transition between Phase II and III), close to the transition between Phase II.a and II.b (see Fig 8C). Simultaneously, observing the emergence of $\beta$ waves in both populations (see Fig 8A and 8B), we identified a relationship between maximum information transfer between inhibitory neurons and $\beta$ waves.

To further elucidate the relationship between $\beta$ waves and information transfer, in Fig 8D, we show that the maximum power spectrum of $\beta$ waves in phase II.a for both populations (x and y axes) correlates with the maximum information transfer in the inhibitory group (see colorbar code). Upon visual inspection, we see that the peak in inhibitory information transfer (with a value larger than 0.25) coincides with the region where $\beta$ waves dominate in both populations (black arrows in Fig 8A, 8C and 8D). However, when $\beta$ waves emerge predominantly only in the excitatory neurons but not in the inhibitory ones—i.e. $0.5 \lessgtr \mu \lessgtr 3$, see also red arrow position in Fig 8A, 8C and 8D –, there is no correspondence with maximum information transfer, indicating the existence of two different informational regimes for $\beta$ wave emergence.

In Fig 9, we focus on data points across different noise levels $\mu$ while maintaining the same $\tau_{rec}$ value ($\tau_{rec} \approx 180$). We observe that when $\beta$ waves have a PSD peak at approximately the same power in both populations (for a noise level $\mu \approx 4.4$ in panels A and B for excitatory and inhibitory respectively), there is a concurrent peak in redundancy (green × symbol data) and information transfer (blue × symbol data) in the inhibitory group (panel B). Although there is also an increase in information transfer in the excitatory group (panel A blue × symbol data), it is less pronounced. For a noise level of $\mu \approx 1.8$, we observe a second PSD peak in $\beta$ waves,

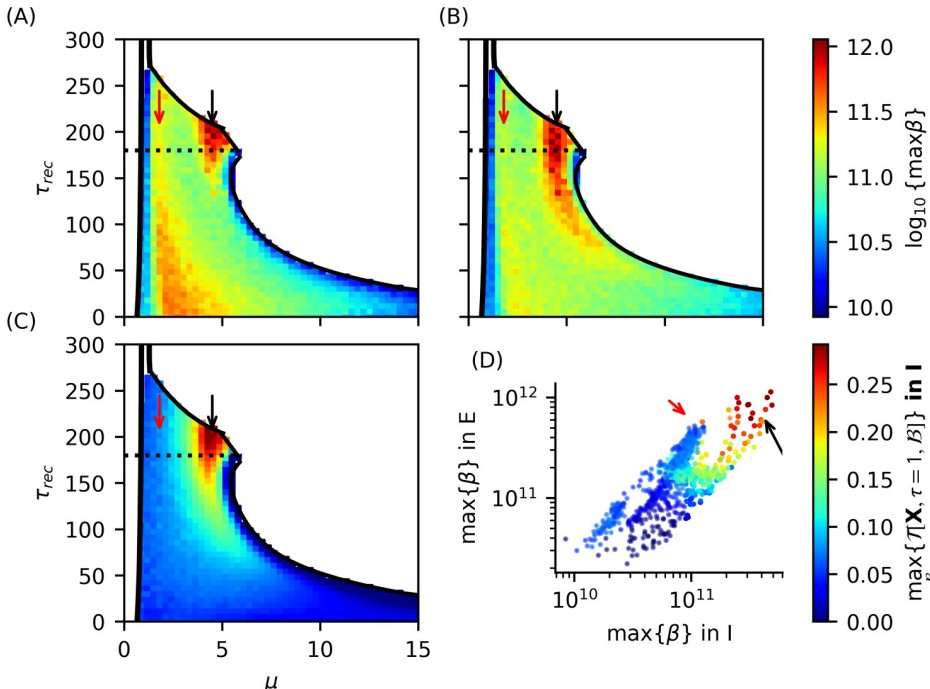

**Fig 8. Relation between β rhythms and information transfer in inhibitory neuron population.** The figure only illustrate regions where β rhythms dominate. (A, B) Color maps show clear regions of maximum of PSD for β rhythms in Phase II.a, for both excitatory (A) and inhibitory (B) neuron populations. White regions correspond to other phases (without dominant β waves) that were ignored. Panel (C) shows the maximum transfer of information (see colorbar code on the right of panel D) between partitions of inhibitory neurons within the Phase II.a. Panel (D) illustrates the transfer of information of the bipartition that maximize information transfer in the inhibitory group. Maximum information transfer increases with increasing power of β waves in both neuron populations (region indicated by black arrow), showing that in our system, β waves are related with information transfer between inter-neurons. Panels (A, C, D) depict, however, that there are points where we see a high PSD in β, mainly in the excitatory population (red arrow), that does not relate with an increase in transfer information. Dotted black horizontal line indicates the value of $\tau_{rec}$ and range of μ used in Fig 9.

which coexists with lower frequency rhythms (Fig 9E). This second peak shows also a peak in redundancy (green × symbol data on panel A around $\mu \approx 1.8$) with almost no information transfer (blue × symbol data in panel A around $\mu \approx 1.8$) in the excitatory population, indicating a different information dynamics at play.

In conclusion, we observe that the region where low (δ, θ and α) and β waves coexists (see Fig 9C and 9E), presents an information dynamics clearly distinguishable from the region where β waves dominates alone (see Fig 9D and 9F). In the former, there is a prevalence of redundancy and differentiation (both related to low-order information storage dynamics [58]). Conversely, in the latter, there is no differentiation in excitatory neurons, and maximum transfer occurs in inhibitory neurons. This suggests that in the latter case, the information dynamics is less related to information storage and more to information transfer.

**Dominant δ waves and excitatory information transfer.** In Fig 10 we explore the metastable region between phases II.a and III [24]. Here, dominant δ waves emerge near the second bifurcation line of this region (solid black line). Interestingly, the maximum information transfer coincides with the emergence of δ waves in the excitatory population, indicating a potential functional relationship. However, the emergence of δ waves in the inhibitory population does not correlate with information transfer. To confirm this relationship beyond simple visual

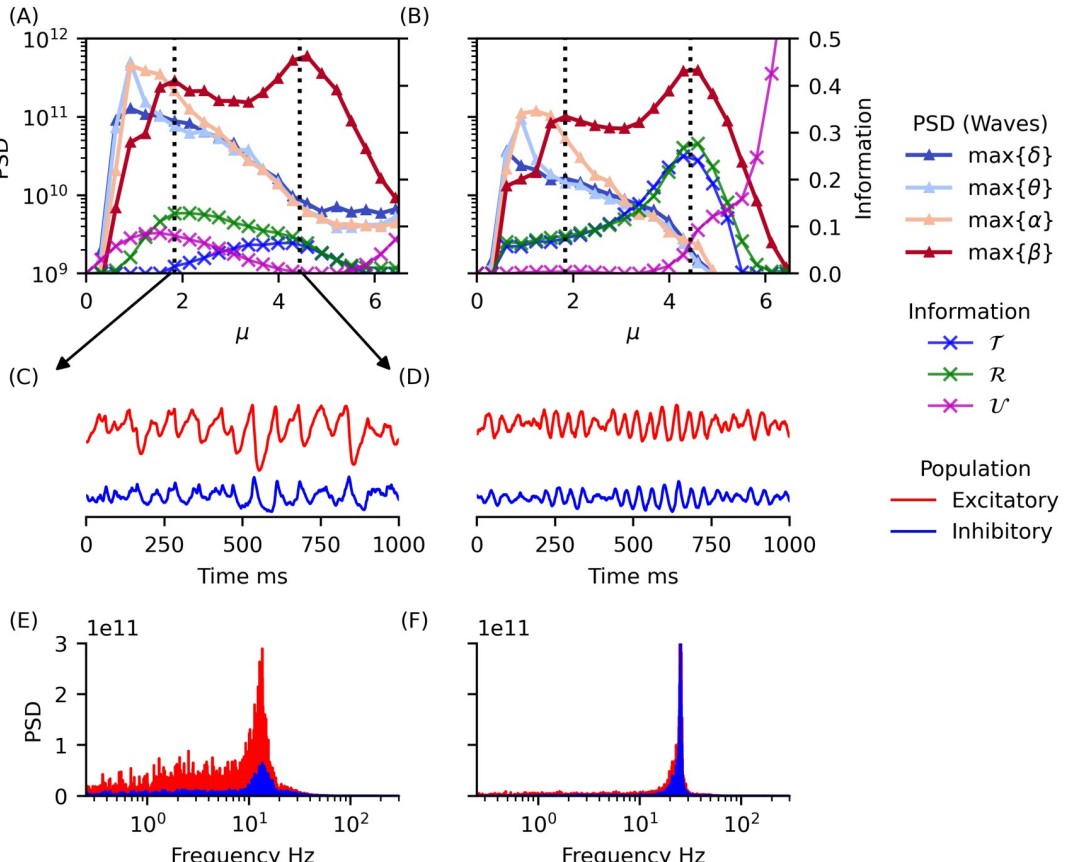

**Fig 9. Comparison of $\delta - \beta$ rhythms coexistence and emergence of dominant $\beta$ rhythm shows different information dynamics profile.** Panels (A, B) depict maximum PSD in $\delta$, $\theta$, $\alpha$ and $\beta$ bands and three information measures (Transfer, Redundancy and Differentiation) within Phase II.a for fixed $\tau_{rec} \approx 180$. (A) Excitatory and (B) Inhibitory neuron populations. As observed in Fig 8, information transfer (blue × symbol data), but also Redundancy (green × symbol data) in inhibitory group are maximum for the same $\mu$ that maximizes PSD for $\beta$ rhythms in both populations (vertical dotted line in $\mu \approx 4.5$). The inhibitory differentiation (magenta × symbol data curve) increases considerable close to the discontinous transition between phase II.b and III (see 7C bottom). Here, we observe the beginning of this behaviour, which are not related to emerging waves in phase II.a. We also observe a first peak of PSD of $\beta$ waves coexisting with slow waves—cf. panel (E) red PSD—that does not relate with a peak in information transfer (vertical dotted line for $\mu \approx 1.8$ in panel (B)). In this "waves coexistence" regime we observe a relatively small peak in Redundancy in excitatory neurons but no information transfer (see green × symbol curve and blue × symbol curve on panel (B)). Panels (C, D, E, F) illustrate the averaged membrane potential fluctuations and the corresponding PSDs for a group of excitatory and inhibitory neuron populations for $\mu \approx 1.8$ (panels C and E) and $\mu \approx 4.5$ (panels D and F). The membrane fluctuations features are clearly different and, while in (C) we have a rhythm with multiple frequency waves including slow and fast components (i.e. $\delta - \beta$ waves coexistence), in (D) we observe a clear dominant $\beta$ rhythm regime.

inspection of Fig 10A and 10C, we examine the dispersion plot of maximum inhibitory and excitatory $\delta$ power and maximum information transfer (colorbar code of plotted data points) in Fig 10D. Here, we observe that in the meta-stable region, the maximum information transfer between excitatory neurons indeed increases with the power of excitatory $\delta$ waves. While inhibitory $\delta$ waves do not exhibit a clear relationship with excitatory information transfer, unlike $\beta$ waves, where we have seen that their emergence in both populations correlates with inhibitory information transfer. These findings highlight the inherent complexity of wave emergence and informational properties in neuronal populations.

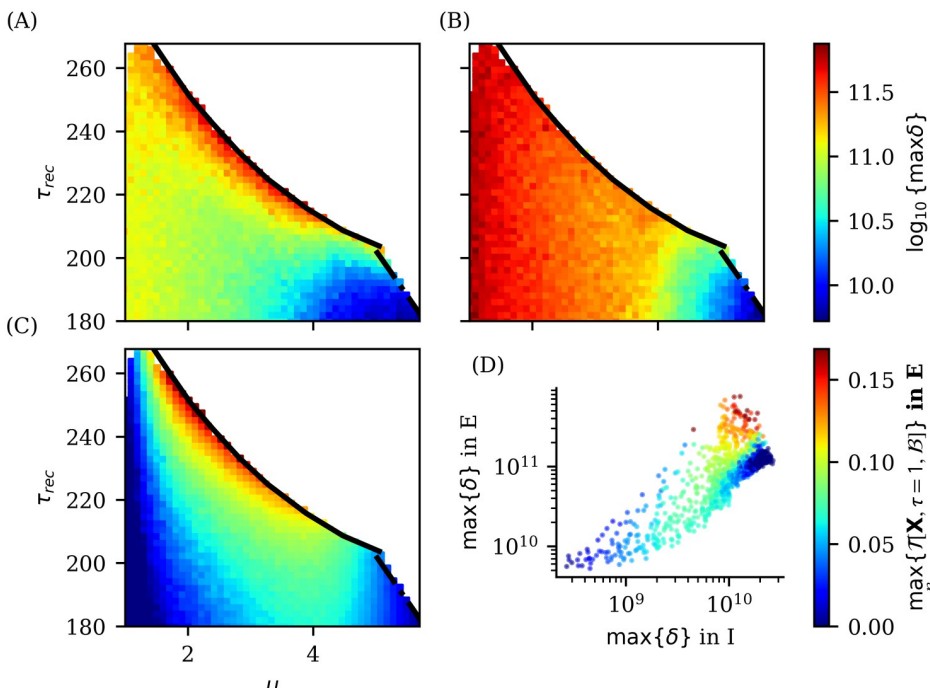

**Fig 10. Relation between $\delta$ rhythm emergence in the meta-stable region and information transfer in excitatory neuron population.** Color maps in panels (A,B) show maximum PSD of $\delta$ rhythm in meta-stable region between II.a and III for excitatory (A) and inhibitory (B) neuronal populations. Panel (C) shows the maximum transfer of information between groups of excitatory neurons. White regions in panels (A-C) are part of the parameter space that we are not taking into account in the present analysis, as they belong to other phases. Panel (D) depicts the information transfer $\mathcal{T}$ of a bipartition that maximizes information transfer in excitatory populations. Maximum information transfer increases with increasing power of the emerging $\delta$ waves in the excitatory population, while shows no clear correlation with $\delta$ waves in inhibitory population.

In summary, we observed an intriguing interplay between the existence of a dynamical phase transition (in this case a discontinuous transition with meta-stability), a specific local information dynamics mode (related with a maximum information transfer between excitatory neurons) and the emergence of a specific meso/macroscopic phenomenon (the generation of $\delta$ waves). Information dynamics provide then insights into the potential functions of the system, while the occurrence of meta-stability is associated to the possibility of two distinct neuronal activity states (up and down). Additionally, the emergence of specific waves, such as $\delta$ waves, can serve as a macroscopic measure to experimentally identify this regime, allowing us to test hypotheses and establish connections with neuroscience literature concerning the functional significance of this $\delta$ brain rhythm.

**Information storage as an ubiquitous phenomenon related to wave emergence and coexistence.** In Fig 11 we present the values of information storage in the phase II.a following the storage measure defined in Eq (18). We observe that in the excitatory population, information storage is ubiquitous to the coexistence of different rhythms (Fig 11A). Note, for example, that for noise levels $1 < \mu < 4$, where we identified the coexistence of $\alpha$, $\theta$, and $\beta$ waves, we also observe high values of information storage. However, the maximum storage values occur both in the meta-stable region (where dominant $\delta$ waves were identified) and close to the LAI phase transition where $\beta$ waves dominate the spectrum (see Fig 6A). This suggests, in general, a relation between emerging waves in the excitatory neuron population and an increase of information storage.

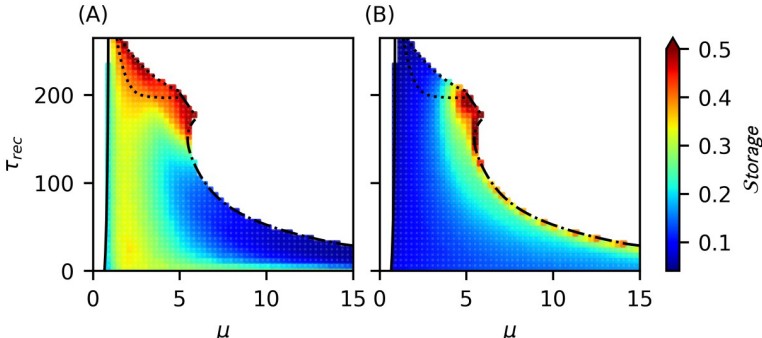

**Fig 11. Relation between information storage in excitatory and inhibitory neuron populations and rhythms emergence.** Information storage on phase II.a in (A) excitatory and (B) inhibitory neuronal groups. We observe higher values of information storage in excitatory population in the meta-stable region where $\delta$ rhythms have their higher PSD values (see Fig 10A), and also close to the LAI phase transition (see Fig 8A) where we observe the higher maximum PSD in the $\beta$ band (as shown previously in Fig 8). Additionally, we can see that information storage in excitatory neurons seems to be an ubiquitous information dynamics for waves emergency in the region where we have coexistence between $\alpha$, $\theta$ and $\beta$ rhythms (as shown when $1 \lesssim \mu \lesssim 4$ in Fig 6A), where also there is also an increase in redundancy in excitatory neurons (see Fig 7B top). This indicates that, in phase II.a, high level of information storage in excitatory neuronal population is related with both, a clear rhythm emergence with strong power (as occurs for $\delta$ and $\beta$ in the regions indicated) and coexistence of low frequency waves. Implications of this result will be discussed later in the text.

On the other hand, the inspection of the Fig 11B shows that the inhibitory population exhibits higher storage values only close to the LAI phase transition, with maximum storage values occurring close to the meta-stable region, where we also observe the strongest PSD in $\beta$ band (see Fig 8B). The behavior of our storage measure in both neuronal population indicates once more that inter-neurons dynamics have inherently distinct functional roles in the system with respect to excitatory population dynamics. As a consequence, a rhythm generated in such inhibitory neuronal population possess different functional properties with respect to the same rhythm in the excitatory neuronal population. This fact impedes in practice to establish a simple, general connection between particular rhythms and circuit functions; we must at least differentiate in which population the rhythms are being generated. We will elaborate further on this point in the Discussion section.

**High-frequency oscillations emerge in two opposing information dynamics regimes.** As shown previously, in phase II.b we observe the emergence of high-frequency oscillations (HFOs) in the range (140 to 190 Hz). These rhythms exhibit two different regimes corresponding to distinct regions: one where rhythms emerge in both neuronal populations with frequencies around 170–190 Hz, and another where rhythms emerge only in the excitatory population at lower frequencies (as shown in top panels of Fig 12. A detailed description of these rhythm properties is provided in the S1 Appendix. We demonstrate that when HFOs emerge dominantly in both populations, they exhibit greater amplitudes and the same frequency. In contrast, when they dominate only in the excitatory population, inhibitory oscillations have higher frequencies but negligible amplitudes (see Figs E and F in S1 Appendix). The emergence of HFOs cannot be explained solely by the high neuronal activity $\rho$ present in this phase. While the temporal variance of inhibitory activity $\rho^I$ correlates with the emergence of $\gamma_{fast}$ in the excitatory population, the reverse is not true (see Fig G in S1 Appendix). These results suggest that simply observing the statistics of a global order parameter cannot fully explain the emergence of these rhythms, indicating a more complex underlying mechanism.

To deeper explore into the complexity observed in HFO emergence in our system, we examine the information dynamics of the neuronal system in phase II.b. Initially, we calculate

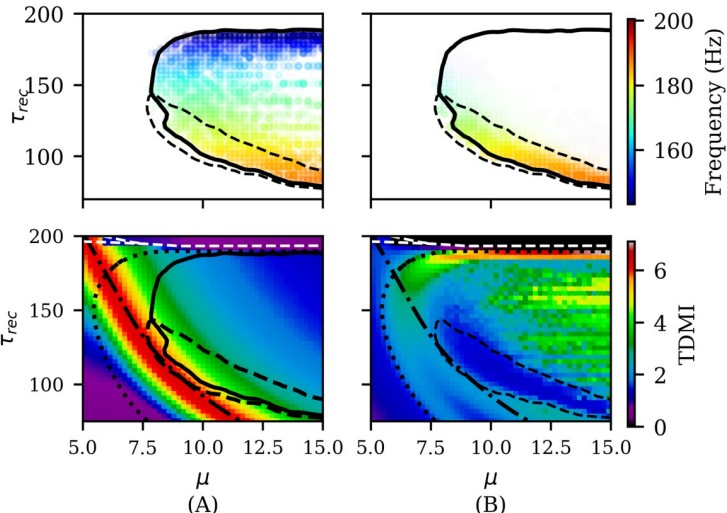

**Fig 12. Time-delayed mutual information and the emergence of fast $\gamma$ rhythms.** (Top) PSD peak frequency of oscillations in excitatory (A) and inhibitory (B) neuron groups, where color opacity is directly proportional to the power spectrum density. (Bottom) Time-delayed mutual information in (A) excitatory and (B) inhibitory neuron groups. The solid black line indicates the region where high frequency excitatory oscillations $\gamma$ emerge (see Fig 6A). The dashed black line indicates the region where inhibitory fast $\gamma$ oscillations with PSD $>1 \times 10^9$ coexist with excitatory oscillations with PSD $>1 \times 10^{11}$ (see Fig 6B). Panel (B) bottom depicts that there is a clear relationship between lower mutual information and inhibitory fast $\gamma$ oscillations. Lower mutual information indicates that the system is less predictable, which means knowing pass states gives less information about the future of the system and vice versa. Dotted and dashed-dotted black lines indicate the line of maximum variance of the inhibitory and excitatory neuron states $X$, respectively.

the time-delayed mutual information (TDMI) for each neuron group (12E and 9I) across the parameter space within the region of interest. The results, depicted in Fig 12, feature solid and dashed black curves denoting regions where excitatory and inhibitory $\gamma_{fast}$ dominate, respectively (A and B top panels). We note that high values of TDMI in the excitatory neuron group does not correlate with regions where excitatory $\gamma_{fast}$ rhythms emerge (see bottom A panel). Conversely, the area where inhibitory $\gamma_{fast}$ waves prevail precisely corresponds to a zone of low TDMI in the inhibitory population (blue region enclosed by the black dashed line at the bottom of 12B). While other regions of low TDMI exist, they lie outside the domain of high excitatory activity $\rho^E$ (phase II.b), rendering direct comparison with phase II.b inappropriate as they pertain to generally lower neuronal activity levels.

Lower TDMI indicates that the knowledge of a past state provides less information about the future state and vice versa, making the system less predictable or more random. Conversely, identifying a clear region with lower TDMI that correlates with the dominant inhibitory $\gamma_{fast}$ frequency region suggests that, from an informational perspective, these rhythms differ fundamentally from the lower frequency excitatory $\gamma_{fast}$. These findings imply that in our system, higher frequency HFOs that emerge simultaneously and at the same frequency in both neuronal populations exhibit poorer informational properties, and therefore, less functional dynamics. To gain deeper insights, we will explore various information measures associated with excitatory and inhibitory $\gamma_{fast}$ waves.

Excitatory $\gamma_{fast}$ rhythms appear to be correlated with higher redundancy in inhibitory neurons, alongside higher differentiated (see Fig 13C). High redundancy suggests greater robustness to failures, while high differentiation implies specialization, indicating that each partition carries distinct information and performs different operations. Together, these findings

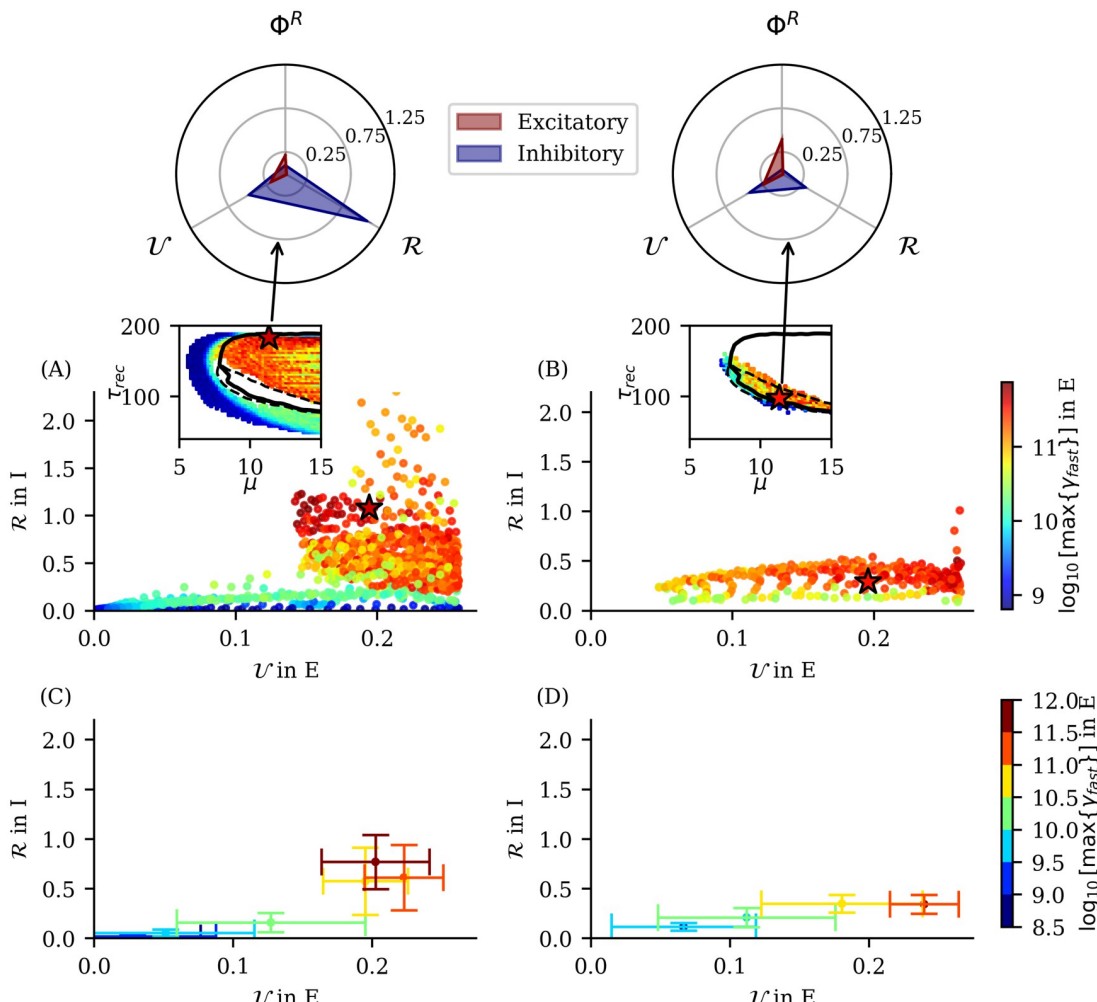

**Fig 13. Two different regimes of $\gamma_{fast}$ rhythms are discriminated by the information dynamics.** Regime 1 (left panels): representative $(\mu, \tau_{rec})$ points in phase II.b, where $\gamma_{fast}$ dominates only in excitatory (E) neuron population. Regime 2 (right panels): representative $(\mu, \tau_{rec})$ points in phase II.b where $\gamma_{fast}$ dominates in both E and inhibitory (I) populations. Panels (A,B) depict the dispersion of the points (shown also in the insets) along the Differentiated information $\mathcal{U}$ in E neurons (x-axis) and Redundant information $\mathcal{R}$ in I neurons (y-axis) plane, for Regime 1 (A) and 2 (B). The inset of Panel (A) shows the maximum PSD of $\gamma_{fast}$ band of E neuronal population in phase II.b, with exception of points corresponding to Regime 2 enclosed by a dashed black line and also appearing in the inset of panel (B). Panels (C,D) shows mean and standard deviation of the information measures values $\mathcal{U}$ in E and $\mathcal{R}$ in I obtained by grouping them into PSD intervals. The intervals are indicated in the discrete colorbar code of panel (D). Regime 1 (A and C) shows that the points with high intensity of excitatory $\gamma_{fast}$ oscillations also have the highest $\mathcal{U}$ and $\mathcal{R}$ observed in the data. Regime 2 (B and D) shows less redundant information in the I population. Panel (C) clearly shows that, in Regime 1, higher $\mathcal{U}$ and $\mathcal{R}$ are associated with dominant high frequency $\gamma$ oscillations (PSD>$10^{10}$). However, panel (D) shows that, in Regime 2, there is almost no increase in redundant information as $\gamma_{fast}$ band maximum PSD increases. Spider graphs on top of panels (A,B) show, for each $\gamma_{fast}$ regime, information dynamic diagrams for two points with high values of maximum PSD in $\gamma_{fast}$ band in the E neuronal population. The representative points were indicated with stars in panels (A,B) and their insets. There, the star's color indicates the maximum PSD of $\gamma_{fast}$ in excitatory population following the colormap code presented in the (A,B) panels.

suggest that, the dynamical regime dominated by excitatory $\gamma_{fast}$ oscillations exhibits a better capacity for robust, specialized computational tasks. On the other hand, there is a clear negative correlation between inhibitory $\gamma_{fast}$ power and TDMI in inhibitory neurons (see Fig 14). Notably, this crucial insights are attainable solely through the current information dynamics

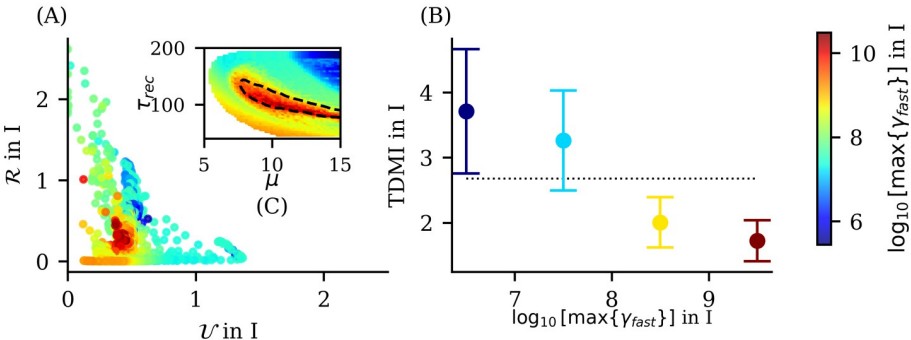

**Fig 14. Emergence of $\gamma_{fast}$ rhythms in I groups and information dynamics.** Colors indicate the level of maximum power density in the $\gamma_{fast}$ band. The data points in (A) correspond to points of high activity $\rho^I$ of the inhibitory population (region II.b) as shown in the inset (C) of the figure. The scatter plot of differentiated ($\mathcal{U}$) and redundant ($\mathcal{R}$) information in inhibitory groups, respectively, shows that the points of highest intensity of inhibitory fast $\gamma$ oscillations have the lowest $\mathcal{U}$ and $\mathcal{R}$ observed in the data. These points are those where excitatory and inhibitory oscillations coexist (see Fig 13F). (B) By grouping data in power spectrum density (PSD) intervals and computing the mean and standard deviation of time-delayed mutual information (TDMI), we observe a clear inverse relation between inhibitory oscillations amplitude and mutual information. The black dashed lines in (B) are the averaged TDMI of all data.

framework, rather than traditional approaches such as order parameter statistics or spectral analysis of neuron membrane fluctuations.

With regard to integrated information in both excitatory and inhibitory groups, we observed that points of maximum excitatory $\gamma_{fast}$ power exhibit lower integrated information levels (Fig 15A), although some exceptions are noted. Conversely, while inhibitory $\gamma_{fast}$ does not appear to be related to the level of integrated information in the excitatory group, it exhibits a clear correlation with lower values of integrated information in the inhibitory group (Fig 15B). The clear association between high frequency oscillations in the inhibitory group and reduced integrated information supports the observation of lower TDMI, as integrated information contributes to the TDMI measure. This behavior agrees with our interpretation that

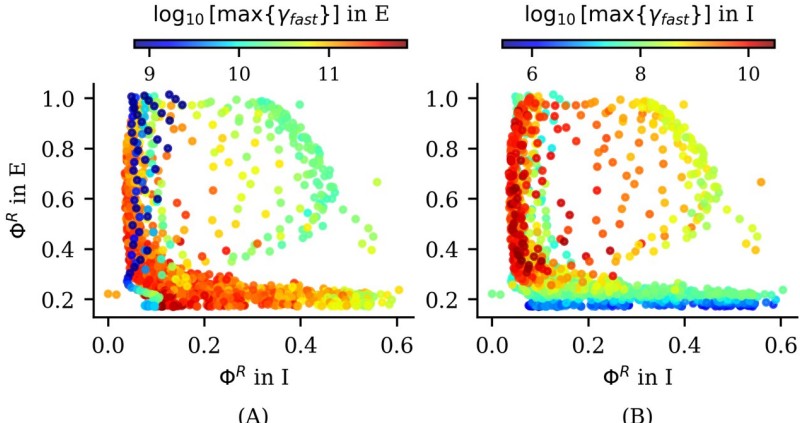

**Fig 15. Emergence of $\gamma_{fast}$ rhythms and the integrated information.** The data points correspond to the high activity $\rho^I$ of the inhibitory population (region II.b) as in Figs 13 and 14. The scatter plot of these points shows the values of the integrated information ($\Phi^R$) in the excitatory and inhibitory groups at each point. The colors indicate the maximum power density level in the (A) excitatory and (B) inhibitory $\gamma_{fast}$ band. Points with high power for $\gamma_{fast}$ appear to concentrate in regions of lower integrated information in inhibitory groups.

inhibitory $\gamma_{fast}$ waves in our system emerge in more random dynamical regimes, where interneurons may be less capable of sustaining higher levels of information processing.

## Discussion

### Model choices and network topological details

The neuronal network model considered in this work follows a specific topology, consistent with approaches established in previous literature. However, the main findings regarding the phase diagram and wave emergence are robust to modifications of this topology. For instance, introducing random connections between excitatory-inhibitory (E-I) and inhibitory-excitatory (I-E) neurons by reconnecting links with a probability $p$ does not alter the observed phase transitions and rhythms, as demonstrated in [24]. However, the emergence of $\gamma$ waves is sensitive to local topological details, resulting in variations in the exact regions of the phase diagram where these rhythms emerge and in their frequency power. This sensitivity is expected, as $\gamma$ waves are closely associated with local circuit dynamics [74–76]; therefore, their emergence depends on specific local connectivity details.

With regard to model parameter choices, as outlined in the Methods section, this model offers a considerable number of parameters that can be explored as control parameters for its dynamics. However, compared to compartmental detailed Hodgkin-Huxley-like conductance models, the number of parameters here is considerably smaller. The selection of values for parameters such as membrane time constants and maximum amplitudes was informed by previous literature on this model [22–24] and allows to reproduce *in silico* actual EEG data features, which simplified our task by providing prior knowledge of relevant parameters and their ranges. Nonetheless, further exploration of the impact of other parameters on system dynamics, such as the maximum amplitudes of synaptic currents between neurons or the proportion of synaptic resources released at each spike, could be the focus of future works.

In our system, the fact of not considering external inputs on the inhibitory neurons designates them the role of interneurons. Through feedback with excitatory neurons, interneurons are responsible for locally controlling neuronal circuits activity, and for processing and transferring information in neuronal circuits [72, 77]. The literature more often assumes that greater diversity in the type of neurons induces a large complexity of the information flow of the neuronal circuit, and at least two types of neurons, parvalbumin (PV) and somatostatin (SST), are often considered to explore this complexity [78]. However, it is surprising to see that even dough in this model all interneurons are the same, a high complex behavior already emerges, as we have observed in the information dynamics (see Fig 7 bottom in phase II.a). This complexity arises primarily due to the introduction of short-term synaptic plasticity, a simple homeostatic mechanism that generates heterogeneity in coupling strength between neurons, facilitating a complex interplay between excitation and inhibition. This conclusion is supported by the observation that when this homeostatic mechanism is disabled ($\tau_{rec} \approx 0$), no complex pattern in the information dynamics is observed, regardless of the noise level $\mu$ considered.

In summary, even though our model considers a very specific topology and many of its dynamical parameters were fixed at specific values, this apparent fine-tuning is not problematic. Previous literature demonstrates the robustness of the main properties of this model for topology modifications and different parameter values. Therefore, since the emergence of rhythms and the different dynamical phases observed are robust, we are pretty sure that our conclusions regarding the information dynamics related to phase transitions and rhythms emergence are also robust, at least for this specific model. Exploring the information dynamics

related to phase transitions and rhythms emergence in different neuronal network models will be the focus of future research.

## Phase transitions and information dynamics

As we observe through information dynamic analysis of the phase diagram, some information measures are clearly sensible to the existence of phase transitions. A concise summary of these observations is provided below:

- Redundancy, information transfer and storage in the excitatory population have peak values in the discontinuous transition between phase II.a and III, in the metastable region.

- Differentiation in inhibitory population shows a peak in the discontinous transition from phase II.b to phase III.

- Integrated information in excitatory population have a peak in the continuous phase transition between I and III, and also in the LAI phase transition between II.a and II.b.

These observations are not surprising, as informational tools like mutual information have demonstrated to be useful in detecting [79, 80] and understanding phase transitions in both low and high-dimensional systems [81]. It's worth noting that a peak in any of our information measures also corresponds to a peak in time-delayed mutual information, as they are components of it. However, what is intriguing is that continuous and discontinuous phase transitions in different neuronal populations exhibit peaks in different information modes. This suggests that, from an informational perspective, they represent fundamentally distinct phenomena.

One of the most intriguing findings from our study is the maximization of excitatory integrated information during the LAI phase transition. This phenomenon may be linked to the influence of critical or quasi-critical regimes [82–84] on information dynamics. Previous research has shown that critical regimes are associated with the maximization of integrated information in simpler models, such as the Ising model [85, 86] and the Kuramoto model [37, 87]. However, in these models, the peak of $\Phi$ is typically associated with a clear second-order phase transition. In contrast, in our case, we observe a closer association with an LAI phase transition and the emergence of a quasi-critical behavior, which is influenced by factors like the presence of inhibition and homeostatic mechanisms, such as short-term plasticity. This transition can be viewed as a disrupted second-order transition, similar to what occurs in sparse neuronal networks when inhibitory elements are present [71].

Another interesting phenomenon that reinforces the idea of $\Phi^R$ as an indicator of critical behaviour is the observed time scale invariance in $\Phi^R$ for the two decades of time delay $\tau$ explored. This could be related to large time correlations in neuronal activity. Large time correlations are expected in systems close or in a critical state [88]. Therefore, $\Phi^R$, being sensitive to the phase transition, could also be sensitive to large time scale correlations.

Moreover, $\Phi^R$ possesses a crucial additional feature as a critical behavior indicator. It successfully detected both the continuous (second-order between phase I and III) and the LAI phase transition (between II.a and II.b) even when measured over a small sample of neurons (only 12 neurons). Conversely, other common criticality identifiers, such as the existence of power laws in the size and duration of neuronal avalanches, could not be accurately measured in such a small population. Therefore, $\Phi^R$ emerges as a potent tool for studying criticality in actual neuronal networks and experimental data.

Finally, it's worth mentioning that $\Phi^R$ was proposed as a universal indicator of complexity [37]. Concurrently, critical phenomena, such as avalanches following power-law distributions, have been associated with the maximization of complexity in actual neuronal systems [89].

Now, our results demonstrate that a well-defined second-order phase transition and an exact critical point are not necessary to maximize integrated information. Instead, it appears that a "relaxed" phase transition, exemplified by the LAI phase transition, is sufficient to exhibit complexity associated with criticality. In this context, our findings could contribute to advancing the critical brain hypothesis. A proper characterization of this LAI phase like transition and it's relation with integrated information would be the scope of future work.

## Rhythms, informational properties, and functions

Through our analysis of information dynamics across the phase diagram, we identified some clear relationships between certain informational properties and rhythms emergence in excitatory and inhibitory neuronal populations. We summarize these as follows:

- $\delta$ rhythms in the excitatory neuronal population are related to the maximization of information transfer in this population (see Fig 10), as well as to information storage (see Fig 11A).

- The coexistence of $\theta$, $\alpha$, and $\beta$ rhythms shows a relation to information storage (see Fig 11A).

- Dominant $\beta$ rhythms in both populations are related to the maximization of information transfer in the inhibitory population (see Fig 8C) and high information storage (see Fig 11).

- $\beta$ and $\gamma_{Low}$ rhythms in the inhibitory population are related to higher integrated information (see Fig 7A bottom) and information storage (see Fig 11B) in phase II.a.

- Dominant $\gamma_{fast}$ rhythms emerging only in the excitatory population show a relation to high redundancy in the inhibitory population while maintaining differentiation in the excitatory population (see Fig 13).

- The emergence of $\gamma_{fast}$ rhythms in both neuronal populations shows poor informational properties in the inhibitory population, with generally low mutual information (see Fig 14).

- $\gamma_{fast}$ rhythms are related to lower integrated information levels in inhibitory neurons (see Fig 15).

In relation to low-frequency waves, mainly $\delta$ rhythms, we observe high information storage and information transfer, primarily in the excitatory population, within parameter space regions where these rhythms dominate. This indicates potential functional roles associated with the dynamics that originate such rhythms. Functions commonly associated with $\delta$ rhythms include memory consolidation and deep NREM sleep [90], a fact which agrees with the high information storage regimes we found. A previous study found a strong correlation between the power spectra of $\delta$ rhythms (0.95–2 Hz) and active information storage (AIS) in the prefrontal cortex using an interesting and novel spectrally-resolved measure [91]. Our results are consistent with these experimental observations. However, we focus on microscopic scale neuronal activity (neuronal raster plots) in a relatively simple *in silico* neuronal medium, whereas the cited study examines local field potential (a mesoscopic measure) in the cortical layers of ferrets. The agreement between our finding and experimental results indicates that the relationship between low-frequency waves and information storage is likely a robust multiscale phenomenon.

Additionally, $\delta$ waves are also linked to working memory [53], which is compatible with both information storage and information transfer. However, we could not link the low-frequency oscillations in the inhibitory population to any specific information dynamics mode. This may be due to the lower power of the emerging $\delta$ rhythms in the inhibitory population (PSD maxima less than $5 \times 10^{10}$) compared to those emerging in the excitatory neuron population (with maxima PSD more than $4 \times 10^{11}$), as shown in Fig A in S1 Appendix. This suggests

that, as inhibitory $\delta$ waves are not a dominant phenomenon in any region of parameter space explored, it is difficult to relate them to specific informational properties.

In phase II.a, $\beta$ rhythms exhibit the clearest dominance both in the extent of the parameter space area where they emerge and in their frequency power. These waves show a strong correlation with informational properties such as information transfer between inhibitory neurons and information storage in the excitatory population. The properties related to $\beta$ emergence suggest strong functional capabilities for the system, as storage, transfer, and processing of information are key components of working memory [92]. In the literature, in fact, working memory is commonly associated with $\beta$ waves [54], particularly in relation to linguistic tasks [93] and verbal information storage [94]. Finding that the emergence of dominant $\beta$ waves in our simple model is related to informational properties similar to those reported in cognitive and behavioral neuroscience literature, is an intriguing fact which deserves further investigation.

In the integrated information $\Phi^R$ phase diagram for inhibitory neurons (see Fig 7A bottom), we observed that, in phase II.a, the region where $\beta$ and $\gamma_{Low}$ rhythms emerge exhibits higher integrated information. This suggests that the network in this region of parameter space relies on the whole system rather than its parts (synergy) to define its time evolution. Synergistic regimes in a neuronal system are typically associated with higher-order and complex cognitive functions [26, 95]. Concurrently, $\beta$ and $\gamma$ waves have historically been linked to "mental activity" [96]. Functions commonly attributed to these rhythms include focusing, action-selection network functions, decision making, and motor planning—activities that require heightened states of awareness [96, 97].

Related to phase II.b and the emergence of $\gamma_{fast}$ rhythms, in the region of high inhibitory redundancy, dominant oscillations have a frequency of around 145 Hz (see Fig 12A top and Fig F in S1 Appendix). This frequency falls within the $\gamma$ frequency range, which is often associated with information processing in neuronal circuits [76]. Although the precise cognitive functions and mechanisms of these oscillations remain subjects of debate [51], the information dynamics properties observed in our study—excitatory differentiation combined with higher interneuron redundancy—suggest that, in this regime, our model is better suited for robust parallel information processing, akin to what is expected in biological neuronal networks.

Higher frequencies (>250 Hz) are more often associated with pathological brain activity [12, 18], with some exceptions. In our model, the $\gamma_{fast}$ frequencies that exhibit less functional information dynamics are around 190 Hz (see Fig 12B top and Fig F in S1 Appendix), closer to the upper limits typically associated with physiological high-frequency oscillations. Therefore, this result is in agreement with the common understanding related to HFOs: higher frequencies are more frequently linked to pathological states, or in our case, to more random interneuron dynamics.

Our results are especially intriguing because the system we are studying here is, a priori, too simple and not particularly designed (fine-tuned) to capture any specific properties of actual biological neuronal networks. Even more intriguing is the fact that the region where excitatory and inhibitory $\gamma_{fast}$ oscillations coexist and share the same frequency—almost 200 Hz—is also the region of phase II.b where mutual information is minimal (as shown in Figs 13 and 14). This suggests that this "synchronization" of excitatory and inhibitory population frequencies is related to a disruption in information flow in the inhibitory population, making the dynamics of this population more unpredictable or random, which is what we hypothesize could be associated to pathological HFOs.

Throughout our results, we observe that the relationship between emerging rhythms and informational properties depends on the neuronal population in question. It is well known that different neuronal populations have distinct dynamical properties and functional roles

[98, 99]. Therefore, it is expected that excitatory and inhibitory groups of neurons, with their different individual dynamics—including, e.g., synaptic receptors operating at different time scales—will exhibit different informational properties when functioning as a group. Moreover, recent experimental results show that even the same neuronal group can perform different primitive computations (e.g., carrying, storing, transferring information) depending on the dynamical regime [100]. Similarly, our results demonstrate that excitatory and inhibitory populations can exhibit a wide range of informational properties depending on dynamical parameters.

However, there is a significant implication of these results: the notion that neuronal population rhythms can be straightforwardly related to neuronal functions is challenged. Our findings indicate that such rhythms cannot be robustly associated to specific functions without first identifying the neuronal population source of the rhythms. While it is possible to disentangle subpopulational sources of signals in mesoscopic data such as local field potentials [101], this is not the case for macroscopic signals such as EEG recordings. Therefore, our results suggest an upper bound to the degree of specificity with which one can relate function and macroscopic neuronal rhythms.

To some extent, these results help to understand why, even within specific brain regions, the same waves could be associated with a variety of functions [102]. For example, $\beta$ waves are related to sensorimotor control and motor preparation, but also to top-down attention and working memory allocation [54]. Our results suggest that this could be a consequence of different populations operating in different information dynamics regimes while exhibiting the same collective oscillations.

It's important to clarify that none of our results should be considered conclusive regarding the relationship between information dynamics and rhythms emergence. This is because we explored a specific model that, as common sense dictates, cannot be considered a universal description of neuronal systems. However, two key points merit attention. First, we were able, from a microscopic model, to identify specific modes of information dynamics, such as peaks in information transfer, storage, or differentiated information, that correlate with the emergence of specific rhythms at meso/macroscopic scale. Second, these modes appear to align well with the functional properties typically associated with these rhythms in the neuroscience literature. This finding is, at the very least, intriguing and warrants further exploration in experimental data and more detailed microscopic models.

## Limitations of our analysis from an experimental perspective

Among the various relationships we observed *in silico* between emerging rhythms and information dynamics, we consider the most relevant to be related to high-frequency oscillations (HFOs) and the distinction between pathological and physiological HFOs. While different strategies have been proposed to distinguish between pathological and physiological HFOs [21], to our knowledge, no existing references have explicitly related pathological waves with less rich information dynamics. Some previous works often use entropy-based time series analysis in combination with machine learning approaches [103, 104] to identify epileptic features in EEG data, trying to avoid using HFO characteristics as biomarkers. Other works have achieved the same performance as with HFO-based markers [105], while others have shown high performance for this task but were retracted due to data manipulation [106]. Other attempts, also related to signal entropy, propose using EEG complexity measures, first finding a relation between seizures and signal complexity decrease, associated with neuronal activity synchronization [107]. However, more recent findings suggest that seizures could be related to an early increase in signal complexity [108], exemplifying the challenges in clearly defining

pathological biomarkers for seizures. In this context, we consider that informational approaches are still poorly explored in the literature.

Our results related to HFOs leads us to propose the following hypothesis: pathological HFOs, typically associated with epileptic seizures, will exhibit less rich information dynamics characterized by a decrease in mutual information between different parts of the neuronal network. In contrast, physiological HFOs maintain high mutual information along with high differentiation and redundancy.

To test this hypothesis in experiments, we propose two paths:

- Microscopic (high-resolution) data analysis: The first approach involves obtaining high spatial and temporal resolution data of neuronal activity in regions where HFOs emerge, in both pathological and physiological conditions. We would then apply the same analysis used in our model to this data. For our model analysis, we utilized large time series ($2 \times 10^5$ time bins) of spike trains from a small group of neurons. Recent technical developments, such as neuropixel probes, now allow access to large-scale single-cell resolution neuronal activity [109]. However, acquiring data with such detailed resolution remains challenging from an experimental standpoint. Also, limited data introduce difficulties in the robust estimation of probability distributions and entropy's measures [110].

- Mesoscopic/Macroscopic variable analysis: The second, more accessible path is to apply information dynamics analysis directly to mesoscopic or macroscopic variables such as local field potentials (LFP), electroencephalography (EEG), intracranial EEG (iEEG), and magnetoencephalography (MEG), provided there is sufficient spatial resolution of the brain region of interest [111]. Using available open data, we can explore other interesting results found in our model in actual data. For instance, using iEEG time series from the temporal lobe during a working memory task [112], we could test the relationship between neuronal activity dominated by $\beta$ waves and information storage and transfer in vivo. Positive results would establish a precedent for using our model as a valuable test bed to explore the relationship between neuronal dynamics, information dynamics, neuronal waves, and functions.

As commented previously, the information dynamics analysis of our model in the present study, follows exclusively the first path. It is important to highlight that Φ-ID have not limitations with respect to the type of variables used and it can be and was applied to both discrete and continuous variables, including blood-oxygen-level-dependent imaging (BOLD) assuming Gaussian random variables, as done in [26]. Therefore, we could have been follow the second path performing our analysis directly over the averaged membrane potential of neuronal groups. However, when dealing with continuous data, the probability distributions and mutual information estimation are trickier and introduce more arbitrariness regarding the choice of estimator methods and estimator parameter, which requires higher level of expertise. Commonly used methods based on k-nearest-neighbor (kNN) methods introduce bias and errors that should be carefully considered [113, 114].

Despite the fact that in the last decade new techniques have been continually developed to address the limitations and bias of continuous-variable mutual information estimation methods [115, 116], we consider the development of discretization methods for continuous variables, such as ordered patterns [117], a more efficient way of dealing with the above cited limitations [118, 119]. In this case, information dynamics analysis could be applied to discrete variables using simple discrete probability estimators, such as counting bins. This and other questions related to continuous variables in our model will be addressed in a future work.

## Conclusion

In this work, we have studied in depth and exhaustively from both dynamical and informational perspectives a neuronal network model to generate *in silico* EEG-like signals. This has allowed us to expand our understanding of the mechanisms involved in the generation of such signals and the complex emergent behavior associated with them. The dynamical approach revealed different phase transitions, most of which were previously described in the literature [23, 24]. However, in the present work, we also identified what appears to be a low-activity intermediate (LAI) phase for both excitatory and inhibitory neuronal populations, with a transition to a high-activity phase, which was not previously reported in the aforementioned works. Additionally, we have studied the main features of these phases and the complex interplay between excitatory (E) and inhibitory (I) neuron populations which is responsible for the emergence of such phases.

More precisely, through spectral analysis of the averaged membrane potential of groups of excitatory and inhibitory neurons, we identified the regions in the considered parameter space —i.e. noise level $\mu$ and synaptic resource recovery time $\tau_{rec}$—where different rhythms ($\delta$, $\theta$, $\alpha$, $\beta$ and $\gamma$) emerge. More importantly, we find that: high frequency oscillations (HFO) emerge in a phase diagram region of high neuronal activity characterized by $\mu > 7$ and $100 < \tau_{rec} < 200$ ms, which have not been explored in previous literature. On the other hand, low frequency oscillations emerge mainly for a low level noise range ($\mu < 7$), and specifically $\delta$ rhythms emerge dominantly in a meta-stable region. We also reported here that middle frequency rhythms, such as $\beta$ and $\gamma_{Low}$, are clearly dominant with a strong frequency power close to the LAI phase transition.

We identified, moreover, correlations between specific informational properties of excitatory and inhibitory neurons and emerging neuronal rhythms. We found, e.g., that the emergence of $\delta$ rhythms is related to maximization of information transfer and storage in excitatory neuronal populations, while $\beta$ waves show a strong correlation with information transfer in the inhibitory population, and both $\beta$ and $\gamma_{Low}$ rhythms are related with integrated information in the inhibitory neuronal population. Additionally, the coexistence of $\alpha$, $\theta$ and $\beta$ rhythms is related to higher redundant information and storage in the excitatory neuronal population.

The HFOs ($\gamma_{fast}$) shows two regimes with fundamentally different information dynamics: one where both excitatory and inhibitory neuronal populations oscillate at the same high frequency (approximately 190 Hz), and another where excitatory oscillations dominate at a lower frequency of around 145 Hz and no relevant HFOs is observed in the inhibitory neuronal population. The first regime features higher inhibitory redundancy and maintains excitatory differentiated information, suggesting suitability for information processing. In contrast, the second regime, characterized by dominant HFOs in both populations, shows very low mutual information in the inhibitory population, indicating a more random and less predictable system behavior.

In general, we observe that dynamical regimes that have fundamentally different local information dynamics properties in each neuronal population, could generate similar neuronal rhythms at meso/macroscopic scale, suggesting that a straightforward and precise association between rhythms and neuronal functions is not possible, unless the neuronal population responsible for the rhythm can first be identified.

Although our model is simple, it could be very suitable to investigate some functional properties of actual neural systems, such as cortical dynamics. This is mainly due to the rich repertoire of emerging behaviors, both from the dynamical and from the information dynamics perspective which allow us to obtain possible functional insights about the different observed rhythms. The information dynamics measures used to describe local dynamics in combination

with the more mesoscopic spectral measure of neuron membrane potential presents a surprisingly coherent picture of informational properties related to neuronal rhythms and fundamental differences between emerging waves in excitatory and inhibitory populations, which could be further investigated both *in silico* and in experiments.

One of the most intriguing results of the present work is the identification of two distinct HFO regimes from the information dynamics perspective. Differentiating HFOs from experimental data is crucial in the clinical diagnosis of epileptic seizures and the identification of epileptogenic regions. Our model suggests that an informational dynamics approach could be very useful to distinguish between physiological and pathological HFOs. Consequently, future work will focus on testing the hypotheses derived from this study using experimental data related to physiological and pathological HFOs and epileptic seizures.

## Supporting information

**S1 Appendix. Complementary results and discussion.** Contains complementary results and discussion presented in three sections: Power spectrum density across phase diagram, Phase diagram and rhythms and Changes of information measures with increasing time delay. (PDF)

## Acknowledgments

The authors thank Pedro A. M. Mediano for fruitful discussions and comments.

## Author Contributions

**Conceptualization:** Gustavo Menesse, Joaquín J. Torres.

**Data curation:** Gustavo Menesse.

**Investigation:** Gustavo Menesse, Joaquín J. Torres.

**Methodology:** Gustavo Menesse, Joaquín J. Torres.

**Project administration:** Joaquín J. Torres.

**Software:** Gustavo Menesse.

**Supervision:** Joaquín J. Torres.

**Validation:** Joaquín J. Torres.

**Visualization:** Gustavo Menesse.

**Writing – original draft:** Gustavo Menesse.

**Writing – review & editing:** Joaquín J. Torres.

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
