## [Decision Letter · Decision Letter 0]

16 Feb 2024

Dear Mister Mereles Menesse,

Thank you very much for submitting your manuscript "Information dynamics efficiently discriminates high γ-rhythms in EEG brain waves" for consideration at PLOS Computational Biology.

As with all papers reviewed by the journal, your manuscript was reviewed by members of the editorial board and by several independent reviewers. In light of the reviews (below this email), we would like to invite the resubmission of a significantly-revised version that takes into account the reviewers' comments.

We cannot make any decision about publication until we have seen the revised manuscript and your response to the reviewers' comments. Your revised manuscript is also likely to be sent to reviewers for further evaluation.

Sincerely,

Daniele Marinazzo

Section Editor

PLOS Computational Biology

Daniele Marinazzo

Section Editor

PLOS Computational Biology

Reviewer's Responses to Questions

**Comments to the Authors:**

Reviewer #1: The review file is attached as a pdf file

Reviewer #2: This is potentially an interesting work on the intersection of EEG network modelling, the role of oscillatory activity in information processing and application of current method of analysis to shed light on information dynamic in multivariate system. In this paper the authors present the phase diagram of the simulated EEG model to characterize different regimes related to the network activity. Subsequently, the authors show how different rhythms emerge in the phase diagram and provide an information dynamic analysis. High-gamma rhythms emerge in different neuronal populations (excitatory or inhibitory) with different information processing characteristics.

The underlying concept of the manuscript, as presented in the abstract and introduction, specifically pertains to using a model to gain insights into the information processing characteristics of low/high gamma oscillations. This aims to enhance our understanding and ability to distinguish between pathological oscillatory activity and that related to cognitive functions. However, the manuscript devotes only a final part to the role of high-frequency oscillatory activity. Furthermore, the analyses are predominantly descriptive and may be overly influenced by the modelling choices. Thus, the differentiation of functional vs pathological stats seems mostly speculative and not supported by additional experimental data, where new insights from the model analysis can, at least in part, be found in EEG data recordings of patients.

In my view, there are two possible directions for improvement:

1) Re-evaluate the paper's main focus. It seems more that the authors want to provide a comprehensive understanding the role of the modeling parameters in the emergence of different oscillatory dynamic and their information processing capabilities. The role of high-frequency oscillations and their role on functional vs pathological brain states distinction can be only discussed at the end in a 'Future direction' section, but it' s not the highly introduced as the main focus of the paper.

2) Alternatively, streamline the initial sections to focus more narrowly on high-frequency oscillations and think of a possible application to EEG data (with open epilepsy datasets, maybe) where the findings on high-frequency oscillation apply or provide new insights on EEG data. Although the modeling approach offers a level of detail that EEG data alone cannot provide, the authors might consider analyses that establish correspondences between the two. This could involve examining the IID differences (such as redundancy, differentiation, etc.), for example in epilepsy versus control groups, as well as mutual information in both the time and frequency domains. The goal would be to determine if similar conclusions can be drawn from EEG data as those inferred from the model.

I can not recommend the paper for publication without this and the following substantial revisions.

Major comments:

The paper requires significant enhancement in terms of writing quality, grammar, and the clear exposition of concepts. The minor comments I provided address only a few issues, but a comprehensive review is needed for improvements.

1. Did the authors assess the robustness of their results by altering the E-I and I-E connections in the networks? For example, by replacing an I-E connection within a circle with a randomly positioned neuron outside of that circle.

2. In Figure 6, do the white areas on the phase diagram represent regions driven solely by noise, without a distinct PSD maximum? A brief explanation in the figure caption would be helpful.

3. Is the coexistence of low-frequency (delta-theta) and high-frequency (gamma) oscillations in the phase diagram related to the low-high frequency coupling often observed in EEG data? Could the model provide new insights in terms of information dynamics?

4. Do the authors have an explanation for the high differentiation information observed in region III of the phase diagram, which appears prominent only at specific ranges of u noise levels? (Refer to Figure 7 and mainly S7).

5. The estimation of Φ-ID in continuous data, without relying on kNN methods, is possible (as shown in Luppi, Mediano et al., 2022, Nature Neuroscience, and Mediano et al., 2021, ArXiv:2109.13186). Why didn’t the authors apply these measures to analyse average membrane potential (LFP-like) in continuous data? This could be discussed in the 'Limitations of our analysis' section."

6. In the 'Limitations of our analysis' section, the authors should include references related to pathological HFOs and the less rich information dynamics and mutual information.

7. All elements depicted in the figures should be explained in their captions. For example, the significance of the black arrows in Figure 6.

8. The supplementary material contains subsection 2.1, 'Excitatory and inhibitory activity', and Figure S5, which appear to be exact duplicates of content from the main text.

Minor comments:

In the following I reported some of the issues with the writing. Please revise the all manuscript since it was impossible to track all grammar and syntax problems.

The title point to EEG brain waves, the paper use only a model maybe change to -EEG neural network model

Abstract

- NN ? Does it mean Neural network ?

- integrate-and-fire

- “However, in regions where high gamma … we observe generally lower ..”

Author summary:

-“i.e. the recording of ..”

-“where EEG-like signals emerge within a neural population”

-“Through this approach , we explore ...”

Improve writing of the Abstract and Author summary

Line 9: is loosely

Line 10: relaxation

Line 82- PID framework is not restricted to only 3 sources. Also PID is multivariate the difference with IID is that PID the does not allow to decompose multiple targets.

Be consistent with the verb tense in the rest of the Introduction

Line 98: “which has not been explored ..”

Line 145: “which provides a”

Line 154 “it can be applied to decompose”

In general I find “give us” too formal.

Line 223: “expressed”

Line 251-254 split sentence

Line 258 “in inducing” ?

Line 261 “Using a simplified approach”

Line 266: “This model aims to capture the essentials of the cerebral cortex

where it is reported that excitatory neurons occur almost four times more than inhibitory ones.”

Line 268: “To couple”

Line 368: “.., making the observation of active information dynamics between distant neurons less probable;”

Line 335: Wrong Figure 3a, maybe Figure 4a ??

Line 384: “attention on Γ waves”. Is Γ symbol introduced already?

Line 388: Finally, we focus (remove and)

Line 689 write always High or H

Figure 6. colored stars ? not starts

Wrong figure reference in Supplementary material. Many figures show Figure ??

**Have the authors made all data and (if applicable) computational code underlying the findings in their manuscript fully available?**

Reviewer #1: **No: **I did not find in the manuscript the Data and Code availabiltiy section.

Reviewer #2: **No: **The authors did not provide the code for the modeling and the analysis.

PLOS authors have the option to publish the peer review history of their article (what does this mean?). If published, this will include your full peer review and any attached files.

Reviewer #1: No

Reviewer #2: No
---

## [Decision Letter · Decision Letter 1]

10 Jul 2024

Dear Mister Mereles Menesse,

Thank you very much for submitting your manuscript "Information dynamics of in silico EEG Brain Waves: Insights on oscillations and functions" for consideration at PLOS Computational Biology. As with all papers reviewed by the journal, your manuscript was reviewed by members of the editorial board and by several independent reviewers. The reviewers appreciated the attention to an important topic. Based on the reviews, we are likely to accept this manuscript for publication, providing that you modify the manuscript according to the review recommendations.

Sincerely,

Daniele Marinazzo

Section Editor

PLOS Computational Biology

Daniele Marinazzo

Section Editor

PLOS Computational Biology

Reviewer's Responses to Questions

**Comments to the Authors:**

Reviewer #1: The reviewed manuscript has significantly clarified several open issues that I raised during the first review. However, there are a few minor points which are not clear. I think the final manuscript should try to clarify the following points.

INTRODUCTION

There is an imprecision about the definition of HFOs. There is no such thing as “HFOs in the γ band (30-150 Hz)”. Normally HFO are defined above approximately 80 Hz. To avoid confusion, I would stick to the definitions HFOs given by this paper:

https://www.nature.com/articles/s41467-020-18975-8

See refs 3 and 4 in the above paper for example

The definition of gamma band activity is also not precise. Studies investigating gamma-band neural activity are often considering two types of neural modulations. A narrow-band low-gamma oscillations in the 30–80 Hz range during sensory stimulation (Ray and Maunsell, 2010) and cognitive processes, such as attention (Fries et al., 2001; Bosman et al., 2012) and working memory (Pesaran et al., 2002). And a second type of high-gamma range (from 60 to 150 Hz) activity is observed in invasive (e.g., Brovelli et al., 2005; Crone et al., 2006; Jerbi et al., 2009) and noninvasive (MEG) studies. This second type of gamma band activity is broadband rather than purely oscillatory. I think it would be better to specify which type of gamma activity the manuscript is focusing on and whether it is possible to discern the two.

The introduction still contains the presentation of several results. I would move these paragraphs from the introduction to the results section. Lines 143 - 167 for example.

RESULTS

If the labelling of the different oscillatory bands is modified according to the novel nomenclature (first point above), I would suggest revising the figures.

CODE

The data and code underlying the findings described in their manuscript should be fully available without restriction. The repository is very limited and it could contain some additional script for the analysis and visualisation of the resutls. Maybe some exemplar notebook could help. I leave this point to the authors.

Reviewer #2: minor revision attachment

**Have the authors made all data and (if applicable) computational code underlying the findings in their manuscript fully available?**

Reviewer #1: **No: **Only the core code to similate the model. I asked for a mode detailed code in the revision.

Reviewer #2: Yes

PLOS authors have the option to publish the peer review history of their article (what does this mean?). If published, this will include your full peer review and any attached files.

Reviewer #1: No

Reviewer #2: **Yes: **Pinzuti Edoardo

Figure Files:

Data Requirements:

Reproducibility:

References:

---

## [Editor Report · Decision Letter 2]

26 Jul 2024

Dear Mister Mereles Menesse,

We are pleased to inform you that your manuscript 'Information dynamics of in silico EEG Brain Waves: Insights into oscillations and functions' has been provisionally accepted for publication in PLOS Computational Biology.

Best regards,

Daniele Marinazzo

Section Editor

PLOS Computational Biology

Daniele Marinazzo

Section Editor

PLOS Computational Biology

---

## [Editor Report · Acceptance letter]

26 Aug 2024

PCOMPBIOL-D-23-01956R2 

Information dynamics of in silico EEG Brain Waves: Insights into oscillations and functions

Dear Dr Menesse,

I am pleased to inform you that your manuscript has been formally accepted for publication in PLOS Computational Biology. Your manuscript is now with our production department and you will be notified of the publication date in due course.

With kind regards,

Zsofia Freund
